# A Unifying Relational Perspective on Expressive Lottery Tickets

**Lorenz Kummer** [1 2]  **Samir Moustafa** [1 2 3]  **Anatol Ehrlich** [1 2]  **Franka Bause** [4]
**Marco Nennstiel** [1]  **Przemysław Andrzej Wałęga** [5]  **Nils Morten Kriege** [1 6]

## Abstract

Graph neural networks (GNNs) are widely used, but how parameter sparsity affects the expressivity of relational (RGNNs) and temporal (TGNNs) variants is poorly understood. The Strong Expressive Lottery Ticket Hypothesis (SELTH) posits the existence of sparse GNNs that preserve Weisfeiler-Leman (WL) expressivity on static graphs. We generalize this existence result to a probabilistic statement for multi-relational and temporal domains via the relational WL (RWL). We prove that sufficiently parameterized RGNNs contain sparse subnetworks that maintain 1-RWL expressivity and derive a lower bound on the probability that a random pruning yields such a subnetwork. We show that common TGNNs and cross-graph message passing schemes admit RGNN reformulations such that they inherit these guarantees and, moreover, that the expressivity of a sparse RGNN is connected to its optimization behavior under common update regimes. Experiments instantiate the bound, compare it to empirical probabilities on synthetic data, and study how pre-training expressivity relates to optimization and prediction quality metrics on temporal and molecular benchmarks.

## 1. Introduction

GNNs are powerful models for learning on graph-structured data, where complex objects such as molecules, proteins, or social networks are represented as graphs with feature-enriched nodes and edges. They extend deep learning to domains including finance, social networks, medical data, and chem-/bioinformatics (Lu & Uddin, 2021; Cheung & Moura, 2020; Sun et al., 2021; Gao et al., 2022; Wu et al., 2018; Xiong et al., 2021). A central research direction concerns expressivity, i.e., the ability to distinguish non-isomorphic graphs, typically benchmarked against the WL test, with 1-WL serving as a standard baseline. Foundational advances such as Deep Sets (Zaheer et al., 2017) and Graph Isomorphism Network (GIN) (Xu et al., 2019) established 1-WL as the reference point for evaluating GNN expressivity, spurring efforts to surpass its limitations (Morris et al., 2023). Nonetheless, 1-WL remains sufficient for distinguishing most graphs in widely used benchmarks (Morris et al., 2021; Zopf, 2022), underscoring its practical relevance. These advances, however, do not address the multi-relational setting and focus exclusively on static graphs.

For multi-relational graphs, Barceló et al. (2022) and Huang et al. (2023b) introduce the relational WL (1-RWL) and relate it to the expressivity of RGNNs, as proposed by, e.g., Schlichtkrull et al. (2018); Vashishth et al. (2020), and present $k$-relational GNNs that provably surpass 1-RWL. For temporal graphs, Wałęga & Rawson (2025) analyze temporal message passing (MP) by distinguishing global and local TGNNs, provide RWL characterizations for both, and show that their expressivities are incomparable in general but that local models are more powerful on color-persistent temporal graphs. In contrast, Heeg et al. (2025) study temporal isomorphism via time-respecting paths and propose an RWL test on augmented event graphs, grounding expressivity in the temporal causal topology rather than in the global/local MP distinction.

The Lottery Ticket Hypothesis (LTH), introduced by Frankle & Carbin (2018), posits that large, randomly initialized neural networks contain smaller subnetworks, termed *winning tickets*, that can be trained in isolation to match the full model's prediction quality. LTH has also been extended to GNNs via pruning of both graph structures and GNN parameters, resulting in Graph Lottery Tickets (GLTs) that retain prediction quality while lowering computational demands (Chen et al., 2021; Tsitsulin & Perozzi, 2023). Recently, Kummer et al. (2025b) formalized the link between

---

[1]Faculty of Computer Science, University of Vienna, Vienna, Austria [2]Doctoral School Computer Science, University of Vienna, Vienna, Austria [3]CeMM Research Center for Molecular Medicine of the Austrian Academy of Sciences, Vienna, Austria [4]CISPA Helmholtz Center for Information Security, Saarbrücken, Germany [5]School of Electronic Engineering and Computer Science, Queen Mary University of London, UK [6]Research Network Data Science, University of Vienna, Vienna, Austria. Correspondence to: Lorenz Kummer <lorenz.kummer@univie.ac.at>.

*Proceedings of the 43rd International Conference on Machine Learning*, Seoul, South Korea. PMLR 306, 2026. Copyright 2026 by the author(s).

the LTH and GNN expressivity for static, uni-relational graphs: at a minimum, a winning ticket must allow the model to learn to distinguish any non-isomorphic graphs (or nodes) associated with different targets in the downstream task. Under certain circumstances, a sparsely initialized GNN cannot recover from a loss of expressivity. Kummer et al. (2025b) conclude that preserving expressivity under sparsity is crucial for successfully identifying winning lottery tickets and show that sparse subnetworks exist which preserve 1-WL expressivity, thereby introducing the SELTH. Their analysis is purely existential, restricted to static uni-relational graphs and the corresponding 1-WL variant, and does not extend to temporal or multi-relational settings, which can be parameter heavy (Ehrlich et al., 2026).

To address this gap, we leverage Wałęga & Rawson (2025)'s characterization of local and global MP schemes as instances of 1-RWL on suitable abstractions. Together with the RGNN framework, this enables us to extend the SELTH beyond static graphs, proving that sparsely initialized subnetworks can preserve 1-RWL expressivity in both temporal and multi-relational settings, thereby establishing SELTH as a general principle for expressive and efficient RGNNs and TGNNs. We further apply the framework to cross-graph and hierarchical MP (Fey et al., 2020; Ehrlich et al., 2026).

**Related work.** Building on Frankle & Carbin (2018)'s LTH, Frankle et al. (2019) prune early in training rather than strictly at initialization, and obtain sparse subnetworks without loss of prediction quality or stability. Malach et al. (2020); da Cunha et al. (2022) extend to the Strong Lottery Ticket Hypothesis (SLTH), proving that over-parameterized networks contain subnetworks that achieve high prediction quality without training. Zhang et al. (2021a) explain LTH's generalization benefits: pruning enlarges a convex region of the loss landscape, enabling faster convergence with fewer samples. Zhang et al. (2021b) further validate that pruned subnetworks can match dense networks without repeated prune-retrain cycles.

In GNNs, LTH approaches typically co-prune graph structure and model parameters. Chen et al. (2021); Wang et al. (2022) and Sui et al. (2023) co-prune adjacency matrices and weights to recover GLTs that retain prediction quality and Zhang et al. (2024) automate adaptive pruning to find GLTs without manual tuning. Hui et al. (2023) and (Yuxin et al., 2024) consider co-pruning from an adversarial robustness perspective. Tsitsulin & Perozzi (2023) posit that any graph contains a sparse substructure preserving downstream prediction quality. Yan et al. (2024) leverage SLTH to improve memory efficiency. Kummer et al. (2025b) formally and empirically linked LTH and GNN expressivity via SELTH. In line with Zhang et al. (2021b), Kummer et al. (2025b) show that expressive sparse initializations can speed convergence and improve generalization.

Relating LTH to TGNN expressivity is challenging, as the very notion of isomorphism on temporal graphs is still being clarified. Wałęga & Rawson (2025) lay out two seminal notions of temporal isomorphism: *pointwise* isomorphism (Beddar-Wiesing et al., 2024), which compares snapshots independently, and *timewise* isomorphism, which additionally preserves inter-snapshot time gaps. They formalize global (Longa et al., 2023; Xu et al., 2020; Luo & Li, 2022) and local (Qu et al., 2020; Rossi et al., 2020) MP TGNNs via knowledge graphs and prove tight 1-RWL characterizations. Moreover, they note that event-based temporal graphs can be transformed into the snapshot representation (Gao & Ribeiro, 2022; Longa et al., 2023), concluding that analyzing snapshots subsumes event-based inputs. Hence, our work focuses on this snapshot representation to maximize its generality. Complementing these snapshot-oriented notions, Heeg et al. (2025) introduce *consistent event-graph isomorphism* and prove it is equivalent to static isomorphism on an augmented event graph. Other notions of temporal graph isomorphism are provided by Gao & Ribeiro (2022); Souza et al. (2022). Furthermore Souza et al. (2022) show that TGNNs with injective aggregation/updates are as expressive as a variant of 1-WL that refines colors using time-stamped interactions, whereby walk-aggregating and MP TGNNs are incomparable.

Similarly, the expressivity of cross-graph and hierarchical MP architectures has, to the best of our knowledge, not yet been formally characterized or discussed under sparsity. While Ehrlich et al. (2026) formally show that their proposed architecture (XIMP) subsumes Fey et al. (2020) (HIMP), which, in turn, can distinguish some exemplary molecular graphs indistinguishable to 1-WL-bounded GNNs, neither provides formal guarantees.

**Contributions.** We generalize SELTH to (i) multirelational RGNNs, (ii) local and global TGNNs and (iii) cross-graph and hierarchical MP architectures. We formally show that sparse subnetworks in these architectural classes exist which preserve expressivity under the 1-RWL framework. That is, we prove *Relational SELTH (RSELTH)*, which subsumes the original SELTH, and derive an explicit lower bound on the probability that a random pruning mask yields a maximally expressive subnetwork. We further show that every local or global TGNN can be rewritten as an RGNN, to which RSELTH applies, and that RSELTH similarly applies to certain cross-graph and hierarchical MP patterns, for which our 1-RWL characterization also marks the first formal characterization of their expressivity. For TGNNs, we connect RSELTH with different notions of temporal isomorphism. Finally, we analyze the optimization of sparse RGNNs with the minimal expressivity required to realize all task-relevant distinctions, and show that all surviving parameters remain trainable under small-step first-order methods.

## 2. Preliminaries

A *undirected graph* $G$ is a pair $(V, E)$ of a finite set of *nodes* $V$ and *edges* $E \subseteq \{\{u, v\}, u, v \in V\}$. The node and edge sets of $G$ are denoted by $V(G)$ and $E(G)$, respectively. The *neighborhood* of a node $v \in V(G)$ is $N(v) = \{u \in V(G) \mid \{u, v\} \in E(G)\}$. For a *node labeled* graph $G$, we write $G = (V, E, \lambda)$, where $\lambda : V(G) \to \mathcal{X}$ maps to a label set $\mathcal{X}$. For an *edge labeled* graph $G$, we write $G = (V, E, \tau)$, where $\tau : E \to \mathcal{R}$ maps to a label set $\mathcal{R}$. If $G$ is a *directed graph*, we write $\dot{E} \subseteq V \times V$ and $\dot{\tau} : \dot{E} \to \mathcal{R}$. We define the outgoing and incoming neighborhoods of $v \in V$ as $N^-(v) = \{u \in V \mid (v, u) \in \dot{E}\}$, $N^+(v) = \{u \in V \mid (u, v) \in \dot{E}\}$ and $N^\pm(v) = N^-(v) \cup N^+(v)$. If $G$ is directed and $\mathcal{R}$ is a set of relations (edge types), we refer to $G$ as directed *multi-relational graph* and write $\dot{E}_r = \{(u, v) \in \dot{E} \mid \dot{\tau}(u, v) = r\}$ for $r \in \mathcal{R}$ and $N_r^-(v) = \{u \in V \mid (v, u) \in \dot{E}_r\}$, $N_r^+(v) = \{u \in V \mid (u, v) \in \dot{E}_r\}$ and $N_r^\pm(v) = N_r^-(v) \cup N_r^+(v)$. If $G$ is undirected multi-relational with $\tau : E(G) \to \mathcal{R}$, we write $E_r(G) = \{\{u, v\} \in E(G) \mid \tau(\{u, v\}) = r\}$ and $N_r(v) = \{u \in V(G) \mid \{u, v\} \in E_r(G)\}$. If a bijection $\varphi : V(G) \to V(H)$ with $\{u, v\} \in E(G) \iff \{\varphi(u), \varphi(v)\} \in E(H)$ for all $u, v \in V(G)$ exists, we call the two undirected unlabeled graphs $G$ and $H$ *isomorphic* and write $G \simeq H$. For node labeled graphs $H, G$, the bijection must satisfy $\lambda_H(\varphi(v)) = \lambda_G(v)$ for all $v \in G$. For undirected edge labeled graphs $H, G$, the bijection must satisfy $\tau_H(\{\varphi(u), \varphi(v)\}) = \tau_G(\{u, v\})$ for all $\{u, v\} \in E(G)$ and, for directed edge labeled graphs, $\dot{\tau}_H(\varphi(u), \varphi(v)) = \dot{\tau}_G(u, v)$ for all $(u, v) \in \dot{E}(G)$. For directed multi-relational graphs $H, G$, the bijection must satisfy $(u, v) \in \dot{E}_r(G) \iff (\varphi(u), \varphi(v)) \in \dot{E}_r(H)$, where $\dot{E}_r(G) = \{(u, v) \in \dot{E}(G) \mid \dot{\tau}_G(u, v) = r\}$ for all $r \in \mathcal{R}$. For undirected multi-relational graphs $H, G$, the bijection must satisfy $\{u, v\} \in E_r(G) \iff \{\varphi(u), \varphi(v)\} \in E_r(H)$, where $\dot{E}_r(G) = \{\{u, v\} \in \dot{E}(G) \mid \tau_G(u, v) = r\}$ for all $r \in \mathcal{R}$.

**Relational Weisfeiler-Leman.** Let $G = (V, \dot{E}, \lambda, \dot{\tau})$ be a directed, node-labeled, multi-relational graph with relation set $\mathcal{R}$. The (directed) 1-RWL test maintains a node coloring $c_G^{(k)} : V(G) \to \mathcal{C}$ over iterations $k \geq 0$, initialized as $c_G^{(0)} = \lambda$. Given $c_G^{(k)}$, the next coloring is obtained by aggregating, for each relation $r \in \mathcal{R}$, the multisets of colors in the relation-specific outgoing and incoming neighborhoods $N_r^-(v)$ and $N_r^+(v)$, and then applying an injective hashing scheme to obtain new, unused colors. Concretely, similar to (Huang et al., 2023b), we write the update rule as

$$c_G^{(k+1)}(v) = \Upsilon_3\Big(c_G^{(k)}(v), (\{\!\!\{\, c_G^{(k)}(u) \mid u \in N_r^-(v)\, \}\!\!\})_{r \in \mathcal{R}},$$
$$(\{\!\!\{\, c_G^{(k)}(u) \mid u \in N_r^+(v)\, \}\!\!\})_{r \in \mathcal{R}}\Big),$$

where $\Upsilon_3$ is any fixed injective encoding of its arguments into a new color in $\mathcal{C}$. Let $C_{\mathrm{rwl}}^{(k)}(G) = \{\!\!\{\, c_G^{(k)}(v) \mid v \in V(G)\, \}\!\!\}$ denote the multiset of node colors at iteration $k$. The refinement stabilizes once $|C_{\mathrm{rwl}}^{(k)}(G)| = |C_{\mathrm{rwl}}^{(k+1)}(G)|$. For two input graphs $G, H$, if there exists $k \geq 0$ such that $C_{\mathrm{rwl}}^{(k)}(G) \neq C_{\mathrm{rwl}}^{(k)}(H)$, then $G \not\simeq_{\mathrm{RWL}^{(k)}} H$. Finally, 1-RWL reduces to classical 1-WL if $|\mathcal{R}| = 1$ and $G$ is undirected. Therefore, it can be seen as the relational counterpart to 1-WL, serving as an expressivity baseline for RGNNs.

**Relational GNNs.** Let $G = (V, \dot{E}, \lambda, \dot{\tau})$ be a directed, node-labeled, multi-relational graph with relation set $\mathcal{R}$. We initialize node embeddings from labels via an encoder $\Lambda : \mathcal{X} \to \mathbb{R}^d$, i.e., $h_v^{(0)} = \Lambda(\lambda(v))$. A *relational GNN (RGNN) layer* produces $h^{(l+1)}$ from $h^{(l)}$ at layer $l \leq L$ by first aggregating messages per relation and direction, and then combining the resulting summaries in a permutation-invariant fashion (Barceló et al., 2022).

Representative of common RGNN architectures such as (Schlichtkrull et al., 2018; Vashishth et al., 2020; Huang et al., 2023b), we define an *RGNN layer* that mirrors directed 1-RWL refinement $G = (V, \dot{E}, \lambda, \dot{\tau})$ as

$$m_{r,-}^{(l)}(v) = \Omega_a(\{\!\!\{\Phi_{r,-}^{(l)}(h_u^{(l)}), u \in N_r^-(v)\}\!\!\}), \quad (1)$$
$$m_{r,+}^{(l)}(v) = \Omega_a(\{\!\!\{\Phi_{r,+}^{(l)}(h_u^{(l)}), u \in N_r^+(v)\}\!\!\}) \quad (2)$$

for each $r \in \mathcal{R}$ and

$$h_v^{(l+1)} = \Gamma^{(l)}\Big(\Omega_c(\{\!\!\{h_v^{(l)}\}\!\!\} \uplus \biguplus_{\triangleright \in \{+, -\}} \{\!\!\{m_{r,\triangleright}^{(l)}(v), r \in \mathcal{R}\}\!\!\})\Big). \quad (3)$$

Here, each $\Phi_{r\pm}^{(l)} : \mathbb{R}^d \to \mathbb{R}^d$ and $\Gamma^{(l)} : \mathbb{R}^d \to \mathbb{R}^d$ are learnable functions and $\uplus$ is the *additive multiset union*: if $A, B$ are multisets with multiplicity functions $\mu_A, \mu_B$, then $\mu_{A \uplus B}(x) = \mu_A(x) + \mu_B(x)$. Using $\sum$ as multiset aggregator $\Omega_a$ and combinator $\Omega_c$ and injective $\Phi_r^{(l)}$ and $\Gamma^{(l)}$ (e.g., MLPs with suitable activations and width (Amir et al., 2024; Puthawala et al., 2022; Zaheer et al., 2017)) yields the same distinguishing power as 1-RWL:

*Remark* 2.1. Using $\sum$ aggregation as a permutation-invariant multiset encoder over the elements of a multiset $\mathcal{D}$ alone is injective if the elements of $\mathcal{D}$ are linearly independent (Kummer et al., 2025a). Hence $\Phi_{r,\pm}^{(l)}$ and $\Lambda$ must be chosen accordingly.

Note that normalizations (e.g., degree-based scalings) can be inserted into the sums without changing the 1-RWL upper bound (Barceló et al., 2022). We denote the $L$-layer RGNN architecture stacking layers of the form (1)-(3) as $\Psi^{(L)}$ and write $\Psi_\Theta^{(L)}$ for a concrete parameterization $\Theta$.

**Temporal graphs.** A *temporal graph in the snapshot model* is a finite sequence $TG = ((G_1, t_1), \ldots, (G_n, t_n))$ of undirected node labeled graphs with strictly increasing (real-valued) timestamps $t_1 < \cdots < t_n$, where each snapshot $G_i = (V, E_i, \lambda_i)$ shares the same node set $V$ (node labels may vary with $i$) (Wałęga & Rawson, 2025). For $TG$, *timestamped nodes* are defined as $V^{\text{time}} = \{(v, t_i) \mid v \in V, i \in \{1, \ldots, n\}\} = \text{t-nodes}(TG)$ with time-aware labels $\lambda^{\text{time}}(v, t_i) := \lambda_i(v)$. For a timestamped node $(v, t)$, we follow Souza et al. (2022) and define its *temporal neighborhood* as $N(v, t_j) = \{(u, t_i) \mid \{u, v\} \in E_i, \text{ for some } (G_i, t_i) \in TG \text{ with } t_i \leq t_j\}$. We overload $N(\cdot)$ by context: when the second argument is a time $t_j$, $N(v, t_j)$ denotes the multiset/set of *timestamped* neighbors up to (and including) time $t_j$, whereas $N(v)$ denotes the static neighborhood in a single snapshot. According to Wałęga & Rawson (2025), two temporal graphs $TG_a = ((G_1^a, t_1^a), \ldots, (G_n^a, t_n^a))$ and $TG_b = ((G_1^b, t_1^b), \ldots, (G_m^b, t_m^b))$ are *timewise isomorphic* if $n = m$, $t_{i+1}^a - t_i^a = t_{i+1}^b - t_i^b$ for all $i \in \{1, \ldots, n-1\}$, and there exists a bijection $\phi$ that is a graph isomorphism between $G_i^a$ and $G_i^b$ for every $i \in \{1, \ldots, n\}$. If this is the case, we write $TG_a \simeq_{\text{time}} TG_b$. The timestamped nodes $(v, t_i^a)$ and $(u, t_i^b)$ are timewise isomorphic if $\phi(v) = u$.

**Temporal GNNs.** Let $\delta((u, t_i), (v, t_j)) := t_j - t_i$ denote the real valued time gaps between timestamped nodes and write $\delta_{ij}$ as shorthand. Then, following (Wałęga & Rawson, 2025), for a $TG$ as above, a general TGNN is defined as

$$m_{t_j}^{(l)}(v) = \Omega_a(\{\!\!\{\Phi^{(l)}(\Xi(h_\star^{(l)}, \zeta(\delta_{ij}))), (u, t_i) \in N(v, t_j)\}\!\!\}) \tag{4}$$

whereby $\star = (u, t_i)$ yields *global* and $\star = (u, t_j)$ yields *local* MP, $\zeta$ denotes some mapping of the scalar $\delta_{ij}$ to a scalar or a vector and $\Xi$ denotes a function combining $h_{(u,t_i)}^{(l)}$ or $h_{(u,t_j)}^{(l)}$ with $\zeta(\delta_{ij})$, whereby Wałęga & Rawson (2025) chose concatenation $\|$ for $\Xi$ and $\zeta(\delta_{ij}) = \delta_{ij}$. This followed by a combinator

$$h_{(v,t_j)}^{(l+1)} = \Gamma^{(l)}\Big(\Omega_c(\{\!\!\{h_{v,t_j}^{(l)}\}\!\!\} \uplus \{\!\!\{m_{t_j}^{(l)}(v)\}\!\!\})\Big). \tag{5}$$

By Rossi & Ahmed (2015); Xu et al. (2019), summation $\sum$ or concatenation $\|$ are common choices for $\Omega_a, \Omega_c$. Similar to RGNNs, we denote the $L$-layer TGNN stacking layers of the form (4)-(5) as $\Psi$ or $\Psi_\Theta^{(L)}$. Where ambiguous, we specify whether $\Psi_\Theta^{(L)}$ denotes an RGNN or a TGNN.

**Expressivity of TGNNs.** To bridge the MP schemes of TGNNs with the 1-RWL characterization of expressivity, Wałęga & Rawson (2025) construct from $TG$ two directed, multi-relational knowledge graphs $K_{glob}$ and $K_{loc}$ over the timestamped nodes $V^{\text{time}}$ and a finite *relation set of*

time lags $\mathcal{R} = \{0, 1, \ldots, n-1\}$, whereby the $r \in \mathcal{R}$ is obtained from the real valued time stamps via the order-preserving bijection $\iota : \{t_1, \ldots, t_n\} \to \{1, \ldots, n\}$ with $\iota(t_i) = i$, i.e., $r = \iota(t_j) - \iota(t_i) = j - i$. Then, for $K_{glob}$, the relational edge sets for each $r \in \mathcal{R}$ can be constructed over the directional edge sets $\dot{E}^{glob} = \{(j - i, (v, t_i), (u, t_j)) \mid i \leq j, \{u, v\} \in E_i\} \subseteq \mathcal{R} \times V^{\text{time}} \times V^{\text{time}}$ as $\dot{E}_r^{glob} = \{e \in \dot{E}^{glob} \mid \dot{\tau}^\star(e) = r\}$ with $\dot{\tau}^\star((j - i, (v, t_i), (u, t_j))) = r = j - i$ (with $^\star$ indicating the adaptation to the triplet domain). For $K_{loc}$, they are constructed via $\dot{E}^{loc} = \{(j - i, (v, t_j), (u, t_j)) \mid i \leq j, \{u, v\} \in E_i\} \subseteq \mathcal{R} \times V^{\text{time}} \times V^{\text{time}}$ and $\dot{E}_r^{loc} = \{e \in \dot{E}^{loc} \mid \dot{\tau}^\star(e) = r\}$.

For these $K_{loc}$ and $K_{glob}$, the following trivially follows from Wałęga & Rawson (2025):

**Corollary 2.2.** *Let $TG$ be a temporal graph and let $x = (v, t_i)$, $x' = (u, t_j) \in V^{\text{time}}$. For any $k \in \mathbb{N}$, for any local or global TGNN, and 1-RWL instantiated with the per-relation, per-direction neighborhoods $N_r^+(\cdot), N_r^-(\cdot)$ of the corresponding $K_\star(TG)$, with $\star \in \{\text{glob}, \text{loc}\}$, there exist parameters for that TGNN such that*

$$c_{K_\star(TG)}^{(k)}(x) = c_{K_\star(TG)}^{(k)}(x') \iff h_x^{(k)} = h_{x'}^{(k)}.$$

The proofs of the above and all subsequent theoretical claims are provided in Appendix D.

## 3. Relational Expressivity and Lottery Tickets

We now extend SELTH to RGNNs. We assume that all binary pruning masks $M$ with sparsity ratio $\rho \in (0, 1)$ are constructed such that each entry is an independent Bernoulli random variable, and we write $M \sim^{i.i.d.} \mathcal{B}(1 - \rho)$. Furthermore, we assume a parameter initialization $\Theta_0$ for $\Psi^{(L)}$ where each entry is independently drawn from a continuous, bounded uniform distribution on an interval $[c, d] \in \mathbb{R}$ with $c < d$, and we write $\Theta_0 \sim \mathcal{U}_c^d$. A sparse initialization of $\Psi^{(L)}$ is then given by the Hadamard product $\widehat{\Theta}_0 = M \odot \Theta_0$. We also assume, for all MLPs (e.g., $\Gamma^{(l)}$), a class of real-analytic, injective, continuously differentiable zero-fixing activations $\sigma$ with a nowhere-zero derivative. The assumptions on $\sigma$ are merely technical and simplify the analysis. Extending our results to other activations is straightforward but requires case distinctions. We assume that initial node labels are encoded by an injective map $\Lambda$. Appendix H details all assumptions and limitations.

**Theorem 3.1** (Relational SELTH). *Let $\mathcal{D}$ be any finite collection of finite graphs, where each element of $\mathcal{D}$ is a directed node-labeled multi-relational graph $G = (V, \dot{E}, \lambda, \dot{\tau})$. Let $\Psi^{(L)}$ be a corresponding sufficiently over-parametrized depth-$L$ RGNN. Then, for all $G_a, G_b \in \mathcal{D}$,*

$$G_a \not\simeq_{\text{RWL}^{(L)}} G_b \iff \widehat{\Psi}_{\widehat{\Theta}_0}^{(L)}(G_a) \neq \widehat{\Psi}_{\widehat{\Theta}_0}^{(L)}(G_b)$$

*with a probability at least $\gamma_{\mathrm{RGNN}} > 0$.*

Intuitively, $\gamma_{\mathrm{RGNN}}$ is the probability that *all* branch and combine MLPs remain injective on the finite input sets they witness on $\mathcal{D}$. The proof of Theorem 3.1 (Appendix D) yields an explicit lower bound in terms of: the maximum number $N_{\max}$ of distinct inputs witnessed by any $\Phi^{(l)}_{r\pm}$ or $\Gamma^{(l)}$, the minimum $\ell_0$-separation $s_{\min}$ between such inputs, the minimum hidden width $m_{\min}$ across these MLPs, the number of branches $|\mathcal{B}|$ per layer (tied to the relations and directions), the MLP depth $M$, message-passing depth $L$, and the pruning probability $\rho$. Appendix D derives $\gamma_{\mathrm{RGNN}}$ by bounding per-block non-injectivity on $\mathcal{D}$, union-bounding within each layer, and composing over $L$ layers under independent masks. A compact (slightly looser) bound is

$$\widetilde{\gamma}_{\mathrm{RGNN}} \geq \left([1 - \binom{\widetilde{N}_{\max}}{2}\rho^{\widetilde{s}_{\min}\widetilde{m}_{\min}}]_+\right)^{L(M|\mathcal{B}|+1)},$$

with $[x]_+ := \max\{x,0\}$, where $\widetilde{N}_{\max} := \max\{N_{\max}, N_{\max,\mathrm{comb}}\}$, $\widetilde{s}_{\min} := \min\{s_{\min}, s_{\min,\mathrm{comb}}\}$, and $\widetilde{m}_{\min} := \min\{m_{\min}, m_{\min,\mathrm{comb}}\}$ aggregate worst-case input-count, separation, and width across branch and combine MLPs; the full expression is given in Appendix D. For any target $\gamma_{target} \in (0,1)$, it suffices to choose $\widetilde{m}_{\min}$ large enough so that the (clamped) bound $([1 - \binom{\widetilde{N}_{\max}}{2}\rho^{\widetilde{s}_{\min}\widetilde{m}_{\min}}]_+)^{L(M|\mathcal{B}|+1)}$ exceeds $\gamma_{target}$. Provided $1 - \binom{\widetilde{N}_{\max}}{2}\rho^{\widetilde{s}_{\min}\widetilde{m}_{\min}} > 0$, a *sufficient* width to certify $\widetilde{\gamma}_{\mathrm{RGNN}} \geq \gamma_{target}$ is

$$\widetilde{m}_{\min} \geq \frac{1}{\widetilde{s}_{\min}\ln\rho}\ln\left(\frac{1-(\gamma_{target})^{1/(L(M|\mathcal{B}|+1))}}{\binom{\widetilde{N}_{\max}}{2}}\right).$$

This threshold is obtained by inverting a conservative lower bound and may be pessimistic. We therefore call $\Psi^{(L)}$ *sufficiently overparameterized* if its smallest hidden width $\widetilde{m}_{\min}$ satisfies the above inequality, which certifies $\gamma_{target}$.

Theorem 3.1 is not a restatement of Kummer et al. (2025b)'s SELTH, which is an *existence* result for 1-WL bounded GNNs. Theorem 3.1 gives a *probabilistic* guarantee: under a random sparse initialization, directed 1-RWL distinguishability on *directed multi-relational* graphs is preserved with probability at least $\gamma_{\mathrm{RGNN}} > 0$. This requires $|\mathcal{B}|$ relation-direction branches plus a combine MLP and makes $\gamma_{\mathrm{RGNN}}$ depend on worst-case input statistics across all branch and combine blocks. With $|\mathcal{R}| = 1$ and no directions, Theorem 3.1 reduces to Kummer et al. (2025b)'s SELTH.

**Temporal expressivity and lottery tickets.** To bring RSELTH to TGNNs, we first derive the following graph-level statement from the node-level formulation of Corollary 2.2 to match the setup of Theorem 3.1:

**Lemma 3.2.** *Let $TG_a, TG_b$ be arbitrary elements of a finite collection $\mathcal{D}$ of temporal graphs and $\Psi^{(L)}$ a depth-$L$ local*

*or global TGNN. Then, with $\star \in \{glob, loc\}$ being $\Psi^{(L)}$'s MP paradigm, there exist $\Theta$ for $\Psi^{(L)}_\star$ such that*

$$K_\star(TG_a) \not\simeq_{\mathrm{RWL}^{(L)}} K_\star(TG_b) \iff \tag{6}$$
$$\Psi^{(L)}_{\Theta,\star}(TG_a) \neq \Psi^{(L)}_{\Theta,\star}(TG_b). \tag{7}$$

To show that Theorem 3.1 also applies to TGNNs, we show that Lemma 3.2 also holds under pruning. To achieve this, we consider several different routes. Any global or local TGNN can be rewritten as an RGNN operating on an augmented $\widetilde{K}_\star(TG)$ (either encoding the real valued time gaps $\delta_{ij}$ as edge features or using a refined relation alphabet) to which Theorem 3.1 applies. We discuss the case of a $\widetilde{K}_\star(TG)$ encoding $\delta_{ij}$ as edge features here and the alternative route along its shortcomings in Appendix A.

To this purpose, we construct an RGNN operating on an augmented $\widetilde{K}_\star(TG)$ that is algebraically identical to the corresponding TGNN operating on $TG$: Fix $\star \in \{glob, loc\}$ and a TGNN layer as in Eqs. (4)-(5) with aggregation/combination $\Omega_a, \Omega_c$, message map $\Phi^{(l)}$, combiner $\Gamma^{(l)}$, gap encoder $\zeta$, edge-time gaps $\delta(y,x)$ and a corresponding $K_\star(TG)$. We can realize the same update with an RGNN on $\widetilde{K}_\star(TG)$ by attaching to each directed edge $e = (y \to x)$ the edge feature $\xi_e := \zeta(\delta(y,x))$, and using a shared message map $\Phi^{(l)}_{r,\pm} \equiv \Phi^{(l)}$, writing $\widetilde{\Phi}^{(l)}(h,\xi) := \Phi^{(l)}(\Xi(h,\xi))$, for an appropriate $\Xi$ (e.g., $\|$). With the same $\Omega_a, \Omega_c, \Gamma^{(l)}$ as in the TGNN, the RGNN update at $x$ becomes

$$m_{r,\pm}(x) = \Omega_a(\{\!\{\widetilde{\Phi}^{(l)}(h_y^{(l)}, \xi_{(y\to x)}) \mid y \in N_r^\pm(x)\}\!\}),$$

$$h_x^{(l+1)} = \Gamma^{(l)}\Big(\Omega_c(\{\!\{h_x^{(l)}\}\!\} \uplus \biguplus_{r,\pm}\{\!\{m_{r,\pm}(x))\}\!\})\Big),$$

which by construction is algebraically identical to Eqs. (4)-(5) for the chosen (global/local) neighborhoods:

**Lemma 3.3.** *Fix $\star \in \{glob, loc\}$ and a TGNN layer with maps $(\Omega_a, \Omega_c, \Phi^{(l)}, \Gamma^{(l)}, \zeta, \Xi)$. Instantiate an RGNN on $\widetilde{K}_\star(TG)$ with $\widetilde{\Phi}^{(l)}(h,\xi) := \Phi^{(l)}(\Xi(h,\xi))$, using the same $(\Omega_a, \Omega_c, \Gamma^{(l)})$ and strict weight tying $\Phi^{(l)}_{r,\pm} \equiv \Phi^{(l)}$ across $(r,\pm)$. Then for every TG, all $x \in V^{\mathrm{time}}$ and all $l \geq 0$*

$$h_x^{(l)} \text{ (TGNN on TG)} = \widetilde{h}_x^{(l)} \text{ (RGNN on } \widetilde{K}_\star(TG)).$$

Together with Corollary 2.2, Lemma 3.3 implies that the RGNN operating on $\widetilde{K}_\star(TG)$ has exactly the same node-level expressivity as 1-RWL operating $K_\star(TG)$, which transfers to the graph level by the same argument as in Lemma 3.2. As $\mathcal{D}$ is finite and each $TG \in \mathcal{D}$ is finite, only finitely many time gaps $\delta$ appear as edge features on $\widetilde{K}_\star(TG)$. Hence Theorem 3.1 applies verbatim to this RGNN via the same finite-domain injectivity argument. Moreover, as we tie parameters ($\Phi^{(l)}_{r,\pm} \equiv \Phi^{(l)}$) and

choose $\Omega_a, \Omega_c, \Gamma^{(l)}$ to match the TGNN exactly, parameter matrices correspond one-to-one and pruning masks transfer verbatim from the RGNN back to the TGNN. Consequently, RSELTH, up to minor modifications, also holds for sufficiently overparameterized global and local TGNNs. We state the corresponding TGNN reformulation of Theorem 3.1 in Appendix B, and, for completeness, prove it directly using Eqs. (4)-(5). Moreover, Appendix C extends the result to the node level, matching the node-level focus of Wałęga & Rawson (2025).

We now relate Theorem 3.1 applied to TGNNs to (Wałęga & Rawson, 2025)'s notion of temporal isomorphism, which is taken to the graph level by the following Lemma.

**Lemma 3.4.** *Let* $TG_a = ((G_1^a, t_1^a), \ldots, (G_n^a, t_n^a))$ *and* $TG_b = ((G_1^b, t_1^b), \ldots, (G_n^b, t_n^b))$ *be temporal graphs. Fix* $L \in \mathbb{N}$ *and* $\star \in \{\mathrm{glob}, \mathrm{loc}\}$. *Then for sufficient* $L$:

$$TG_a \simeq_{\text{time}} TG_b \implies K_\star(TG_a) \simeq_{\text{RWL}^{(L)}} K_\star(TG_b).$$

Therefore, combining Lemma 3.4 with Theorem 3.1 via Lemma 3.3, if a sparse parameterization of a (local or global) TGNN distinguishes two temporal graphs, then those graphs are not timewise isomorphic. This links RSELTH to other notions of isomorphism of temporal graphs. By Wałęga & Rawson (2025), (Beddar-Wiesing et al., 2024)'s pointwise isomorphism is less strict than timewise isomorphism. Heeg et al. (2025)'s notion is also more relaxed than timewise isomorphism: they prove it lies between time-aggregated and time-concatenated isomorphism, with the latter being equivalent to the notions proposed by Gao & Ribeiro (2022); Souza et al. (2022). For unlabeled snapshot graphs, Wałęga & Rawson (2025)'s timewise isomorphism coincides with time-concatenated isomorphism, hence any timewise-isomorphic pair is consistent event-graph isomorphic (equivalently, their augmented event graphs are isomorphic). The converse does not hold in general; equivalence with timewise isomorphism holds only under constraints (Heeg et al., 2025).

**Cross-graph MP (XIMP/HIMP).** Let $G = (V, \dot{E}, \lambda, \dot{\tau})$ be a directed, node-labeled, (multi-)relational graph (e.g., a molecular graph) and let $\{T_i = (V_i, \dot{E}_i, \lambda_i, \dot{\tau}_i)\}_{i=1}^n$ be abstractions of $G$ (e.g., junction tree (Jin et al., 2018), extended reduced graph (Stiefl et al., 2006)). Analogous to Ehrlich et al. (2026) (XIMP) and Fey et al. (2020) (HIMP), we encode atom-abstraction incidence by an assignment matrix $S_i \in \{0,1\}^{|V| \times |V_i|}$, where $S_i[v, u] = 1$ iff $v \in V$ contributes to $u \in V_i$. XIMP takes $G$ and $\{T_i, S_i\}$ as input and computes $\Psi_\Theta^{(L)}(G, \{T_i, S_i\})$ to embed $G$. It performs MP on $G$ and each $T_i$ and two-way inter-graph MP both indirectly (I$^2$MP, $G \leftrightarrow T_i$) via $S_i$ and directly (DIMP, $T_i \leftrightarrow T_k$) via the on-the-fly computed abstraction-abstraction incidence matrix $S_{ik} \in 0, 1^{|V_i| \times |V_k|}$, where $S_{ik}[u, w] = 1$

iff there exists $v \in V$ with $S_i[v, u] = S_k[v, w] = 1$. While I$^2$MP and DIMP both employ a single-layer post-aggregation transformation (a linear map followed by a pointwise nonlinearity), XIMP and HIMP can employ arbitrary standard GNNs for local MP on $G$ and $T_i$. Moreover, XIMP uses mean aggregation and HIMP uses concatenation or sum. Hence, to standardize the analysis, we adopt sum aggregation to leverage its multiset discriminative power (Xu et al., 2019) and assume a single-layer post-aggregation transformation for I$^2$MP, DIMP and MP.

We define the *compound node set* $\widetilde{V} := V \uplus \biguplus_{i=1}^n V_i$ and construct a directed multi-relational *compound graph* $G^\oplus = (\widetilde{V}, \dot{E}^\oplus, \widetilde{\lambda}, \dot{\tau}^\oplus)$ by taking the disjoint union of all intra-graph edges and adding *typed* cross-graph edges: (i) *intra* edges $\dot{E}$ on $V$ and $\dot{E}_i$ on $V_i$ retain their original relation labels; (ii) for each $i$ and each $(v, u)$ with $S_i[v, u] = 1$, add $v \to u$ with relation type $\alpha_i$ and $u \to v$ with type $\bar{\alpha}_i$; (iii) for $i \neq k$, add $u \in V_i \to w \in V_k$ with type $\beta_{ik}$ whenever there exists $v \in V$ such that $S_i[v, u] = S_k[v, w] = 1$ and the corresponding reverse edge with type $\bar{\beta}_{ki}$. Up to notation, our definition is the same as Ehrlich et al. (2026), except that edge types are made explicit. As XIMP applies non-linear transformation post-aggregation, we initialize embeddings by $\widetilde{h}_x^{(0)} = \Lambda(\widetilde{\lambda}(x))$ for $x \in \widetilde{V}$ with an injective $\Lambda$ as in the relational case (Section 3) but with the additional constraint that $\Lambda(\mathcal{X}) \subset \mathbb{R}^d$ is linearly independent (Remark 2.1). A single *cross-graph layer* on $G^\oplus$ is then a relational update as in Eqs. (1)-(3), over $\mathcal{R}^\oplus$, the union of all intra-graph relation types and the cross-graph types $\alpha_i, \bar{\alpha}_i, \beta_{ik}, \bar{\beta}_{ki}$. By tying parameters per relation type (i.e., assigning linear $\Phi_r^{(l)}$ to all cross-graph types and one parameter tied $\Phi_r^{(l)}$ to each set of intra-graph relations per $G$ and $T_i$) and using $\Omega_a = \sigma \circ \sum, \Omega_c = \sum$ with $\Gamma^{(l)} = \mathrm{id}$, this single layer is, except DIMP's normalization, by construction algebraically identical to XIMP's combination of MP with I$^2$MP and DIMP. As DIMP's normalization would not change the 1-RWL bound on $G^\oplus$ (Section 2), we omit it in the analysis.

**Corollary 3.5.** *Let* $\mathcal{D}$ *be a finite collection of inputs* $(G, \{T_i, S_i\}_{i=1}^n)$ *and* $G^\oplus$ *be constructed as above for each element of* $\mathcal{D}$. *Then there exists a parameterization of an RGNN on* $G^\oplus$ *such that, for all pairs in* $\mathcal{D}$ *and all* $L \in \mathbb{N}$,

$$G_a^\oplus \not\simeq_{\text{RWL}^{(L)}} G_b^\oplus \iff$$
$$\Psi_\Theta^{(L)}(G_a, \{T_i^a, S_i^a\}) \neq \Psi_\Theta^{(L)}(G_b, \{T_i^b, S_i^b\}). \quad (8)$$

As the reduction to a relational layer on the compound graph $G^\oplus$ yields a one-to-one correspondence with XIMP's parameters by construction, the pruning masks align. It suffices to directly apply Theorem 3.1 to the corresponding RGNN operating on $\{G^\oplus\}$: sparse subnetworks that preserve 1-RWL distinguishing power on $G^\oplus$ transfer to XIMP (and hence HIMP, as setting $n=1$ with $T_1$ the junction tree recovers HIMP's inter-MP), yielding SELTH for cross-graph MP.

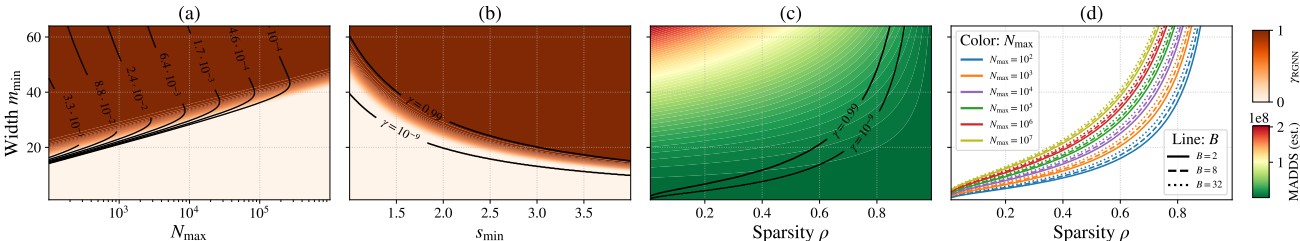

*Figure 1.* **(a)** Landscape of $\gamma_{\mathrm{RGNN}}$ over $N_{\max}$ (log-scale, x) as task complexity proxy and width $m_{\min}$ (y) at fixed sparsity $\rho = 0.70$. Black iso-curves show *efficiency* $= \gamma_{\mathrm{RGNN}}/(\mathrm{MADDs} \cdot 10^{-6})$. **(b)** $\gamma_{\mathrm{RGNN}}$ versus $s_{\min}$ (x) and $m_{\min}$ (y) at $\rho = 0.70$; contours at $\gamma_{\mathrm{RGNN}} = 0.99$ and $10^{-9}$ illustrate that increasing $s_{\min}$ expands the high-$\gamma_{\mathrm{RGNN}}$ region, reducing the width needed to hit the target. **(c)** MADDs cost landscape over $(\rho, m_{\min})$ with $\gamma_{\mathrm{RGNN}} = 0.99$ and $10^{-9}$ frontiers overlaid, showing that achieving $\gamma_{\mathrm{RGNN}} \approx 1$ typically requires either lower pruning (smaller $\rho$) or larger width. **(d)** $\gamma_{\mathrm{RGNN}} = 0.99$ frontiers in the $(\rho, m_{\min})$ plane for multiple $N_{\max}$ (color) and branch counts $B = |\mathcal{B}|$ (line style): larger $N_{\max}$ or $B$ shifts the frontier toward denser or wider regimes. Unless varied per panel: $L = 5$, $B = 4$ (varied only in d), $M = 2$, $s_{\min} = 2$, and $N_{max} = 1000$.

**Optimization of expressive lottery tickets.** The optimization of sparse RGNNs is subtle, so we defer technical details to Appendix E and summarize the main results here. We call a sparse initialization with mask $M$ *optimizable* on a finite dataset $\mathcal{D}$ if, when the output loss gradient is nonzero at least for one element of $\mathcal{D}$, every surviving parameter is gradient-connected to the loss and receives a nonzero update at least once over $\mathcal{D}$. We show for RGNNs that irreducible task-expressive masks (minimal masks whose surviving parameters suffice to distinguish exactly those non-isomorphic graphs in $\mathcal{D}$ that carry different labels) are optimizable in this sense. In contrast, arbitrarily sparsified RGNNs at the same sparsity need not be optimizable: they may both fail to realize label-relevant distinctions and allocate disconnected parameters receiving zero gradients. Finally, we prove that optimizability under irreducible task-expressive masks persists under standard small-step first-order updates for sufficiently small steps sizes.

## 4. Numerical Illustrations and Empirics

Our evaluation has three parts: (i) we instantiate Theorem 3.1's bound $\gamma_{\mathrm{RGNN}}$ to visualize width–sparsity–compute trade-offs as its governing quantities vary (Figure 1); (ii) on a synthetic multi-relational dataset, we instantiate $\gamma_{\mathrm{RGNN}}$ using architecture/dataset-specific input statistics, compare it to an empirical mask success rate $\gamma_{\mathrm{empirical}}$, and relate *pre-training graph separability* (i.e., whether observed graphs receive distinct embeddings) as empirical expressivity proxy to gradient flow and training loss/accuracy under fixed masks as a proxies for optimizability (Figures 2, 3 and 4); (iii) on real temporal and molecular benchmarks, we examine how pre-training separability of pruned models correlates with compression across pruning rates and predicts test performance (Figure 5).

**Setup.** We instantiate Theorem 3.1's $\gamma_{\mathrm{RGNN}}$ to visualize how expressivity and sparsity interact with width and com-

putational cost under varying $N_{\max}, s_{\min}, m_{\min}, |\mathcal{B}|, M, L$ and $\rho$ in Figure 1. We treat $\gamma_{\mathrm{RGNN}}$ as a worst-case probability that a *random* mask preserves the $L$-round 1-RWL expressivity on a finite dataset, not as a predictor of downstream prediction quality. We focus on the RGNN of Section 2 and estimate the multiply-adds operations (MADDs) of an MLP similar to Kummer et al. (2023); see Appendix F.

Figure 2(a) illustrates the tightness of $\gamma_{\mathrm{RGNN}}$. We first generate a synthetic dataset $\mathcal{D}_{syn}$ of 30 multi-relational graphs. Then, we randomly initialize and prune a simple RGNN using $\sum$ for $\Omega_a$, $\Omega_c$ and readout. Next, we compute graph embeddings over $\mathcal{D}_{syn}$. A pruning mask at sparsity $\rho$ is counted as successful iff the resulting embeddings are pairwise distinct across all graphs. Sampling 100 masks for each $\rho$ on a uniform grid of 50 values in $[0.01, 0.99]$ yields an empirical success rate $\gamma_{\mathrm{empirical}}$, which we plot against the mean $\overline{\gamma}_{\mathrm{RGNN}}$ obtained by instantiating $\gamma_{\mathrm{RGNN}}$ on the configuration-specific input statistics of each sample (details Appendix G). Moreover, Figure 2(b) shows how a simple pre-training separability proxy is associated with optimization under pruning. For each $\rho$, we sample 20 masks, apply each mask to a freshly initialized RGNN with a linear classifier, and evaluate the *fraction of separable graphs* before training, i.e., the number of distinct embeddings divided by $|\mathcal{D}_{syn}|$. We then train the model for 10 epochs on a graph-ID classification task over $\mathcal{D}_{syn}$, masking both weights and gradients. We record (i) the final training loss, (ii) the final training accuracy, and (iii) the global gradient-flow (Evci et al., 2022) given by the global $\ell_2$-norm of the gradients, averaged over epochs. We average metrics across trials and report Spearman correlations between pre-training separability and post-training metrics. While Figure 2(b) illustrates a general trend that separability is associated with optimization under pruning over a large interval of pruning rates, it does not fully disentangle expressivity from model size. As subnetworks with lower sparsity naturally retain more parameters, it remains unclear whether the observed improvement in gradient flow, loss, or accuracy comes from preserv-

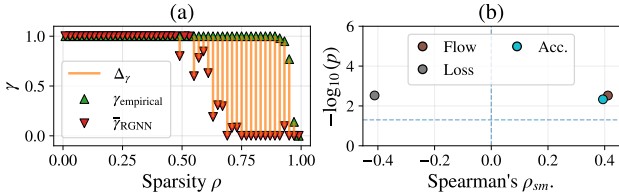 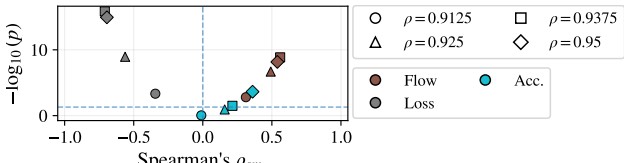

*Figure 2.* **(a)** $\gamma_{\text{empirical}}$ (green) vs. mean $\overline{\gamma}_{\text{RGNN}}$ (red) for randomly sparsely initialized untrained simple RGNNs on $\mathcal{D}_{syn}$. Orange segments visualize $\Delta_\gamma = \gamma_{\text{empirical}} - \overline{\gamma}_{\text{RGNN}}$. **(b)** Spearman correlation between the fraction of separable graphs before training and (i) mean gradient flow, (ii) final training loss, and (iii) final training accuracy. For $\rho$, we average results across trials and compute Spearman's $\rho_{sm}$ and its $p$-value; points plot $(\rho_{sm}, -\log_{10} p)$, with the dashed lines indicating $p = 0.05$ and $\rho_{sm} = 0$.

*Figure 3.* Spearman correlation between the fraction of separable graphs before training and post-training metrics under fixed sparsity (simple RGNN, $\mathcal{D}_{\text{syn}}$). For each $\rho$, we compute Spearman's $\rho_{sm}$ (and $p$-value) across masks between pre-training separability and (i) mean gradient flow, (ii) final training loss, and (iii) final training accuracy. Points plot $(\rho_{sm}, -\log_{10} p)$, dashed lines indicate $\rho_{sm} = 0$ and $p = 0.05$.

ing task-relevant expressivity or simply from having larger subnetworks. Figure 3 hence shows a fixed-sparsity control experiment: using the same RGNN configuration and $\mathcal{D}_{syn}$, for four sparsity levels $\rho \in \{0.9125, 0.925, 0.9375, 0.95\}$, we sampled 100 random masks with identical parameter counts and compared optimization behavior across different pre-training separability levels, using 30 training epochs.

To complement Figure 2(a), in Figure 4, we perform a per-factor study of bound tightness using the same synthetic multi-relational graph generator and the same simple, randomly initialized, untrained RGNN as above. Unless varied, we fix the common hidden width and remaining dataset/model parameters to $(m, |\mathcal{D}_{\text{syn}}|, n, L, |\mathcal{R}|) = (16, 30, 6, 3, 3)$, and sweep $m, n, |\mathcal{D}_{\text{syn}}| \in \{4, 8, 16, 32, 64\}$, and $L, |\mathcal{R}| \in \{2, 4, 6, 8, 10\}$ one at a time. For each configuration, we evaluate 50 uniformly spaced sparsity levels $\rho \in [0.01, 0.99]$. At each $\rho$, we estimate $\gamma_{\text{empirical}}$ and $\gamma_{\text{RGNN}}$ from 10 independently sampled pruning masks and associated statistics, respectively. We summarize tightness by the integrated absolute gap $\int |\gamma_{\text{empirical}}(\rho) - \overline{\gamma}_{\text{RGNN}}(\rho)| \, d\rho \approx \sum_\rho \Delta_\gamma(\rho)$ (see Figure 2(a)), so smaller values indicate closer agreement between $\gamma_{\text{RGNN}}$ and $\gamma_{\text{empirical}}$.

Figure 5(a) relates compression ratio, test performance and separability ratio of the untrained, pruned model, measured (i) at the node level after the final message-passing step for local and global TGNNs on the node-classification benchmark tgbn-trade (Huang et al., 2023a), and (ii) at the graph level for XIMP on one polaris ADMET (HLM) and potency (pIC50 MERS-CoV Mpro) regression benchmark (ASAP Discovery x OpenADMET, 2025a;b) each. Motivated by Figure 2(a), we sweep pruning ratios $\rho$ over 200 uniformly spaced values in $[0.0, 0.995]$, with 5 samples per $\rho$. We train XIMP for 50 epochs and the TGNNs for 10 epochs, using width 16 for all learnable transformations. Figure 5(b) shows the same setup but from a probabilistic perspective. We estimate, for each model-dataset pair, the probability that a pruned model achieves test performance within a small margin of its dense counterpart, conditioned on its

pre-training separability. Separability values are grouped into 25 bins, and for each bin we report $P(\text{WT} \mid \text{Sep.})$, the probability of a model being a winning ticket (WT) [1].

**Results and practical implications.** The numerical behavior of $\gamma_{\text{RGNN}}$ suggests practical design rules for sparse RGNN initializations. Figure 1(d) shows that as $N_{\text{max}}$ and $B$ increase, the width–sparsity trade-off shifts, so larger and more complex benchmarks typically tolerate less aggressive pruning (or require wider layers) to maintain a fixed $\gamma_{\text{RGNN}}$. Moreover, Figure 1(b) shows that increases in $s_{\text{min}}$ (which could be achieved via input encodings (Kummer et al., 2025a) or by avoiding pruning schemes (Deng et al., 2020) inherently zeroing activations) can increase the sparsity attainable at a given width for the same target $\gamma_{\text{RGNN}}$. This also suggests a link to over-smoothing (Chuang et al., 2025; Rusch et al., 2023), which can similarly reduce input separability. The $\gamma_{\text{RGNN}}$ contours in Figure 1(c) highlight a width–pruning trade-off: along a fixed target level (e.g., $\gamma_{\text{RGNN}} \approx 0.99$), width can be traded for sparsity up to moderate $\rho$ while keeping the approximate MADDs budget roughly constant. At high sparsity, however, the width required to maintain the target makes cost grow exponentially. Consistently, Figure 1(a) shows diminishing cost-efficiency from further widening at fixed benchmark complexity and sparsity as $\gamma_{\text{RGNN}} \to 1$. Figures 1(a)–(c) show that the band between $\gamma_{\text{RGNN}} = 10^{-9}$ and $\gamma_{\text{RGNN}} = 0.99$ is narrow, suggesting that most configurations are either severely underparameterized or inefficiently overparameterized. Only a small region supports a cost-effective, highly expressive subnetwork, where careful tuning of width and sparsity relative to dataset and architectural properties affects $\gamma_{\text{RGNN}}$.

Figure 2(a) compares mean $\overline{\gamma}_{\text{RGNN}}$ to $\gamma_{\text{empirical}}$. The bound is tight when expressivity holds with high probability (roughly $\rho \lesssim 0.65$ here), becomes more conservative in the transition region where expressivity fails (approx.

---

[1]Code for reproducing results and figures is available at GitHub: https://github.com/lorenz0890/relational_selth

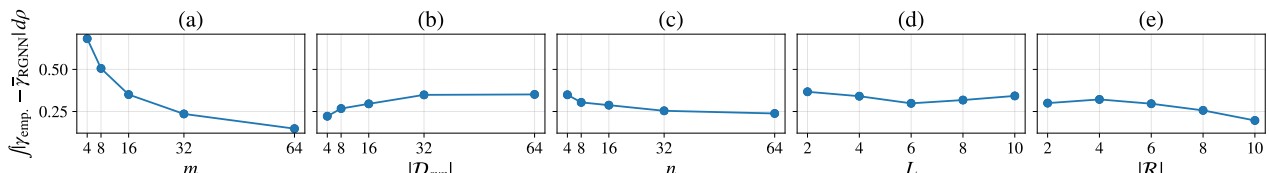

*Figure 4.* Tightness of $\gamma_{\mathrm{RGNN}}$ as model and dataset characteristics vary. Each panel reports the absolute area between $\gamma_{\mathrm{empirical}}$ and mean $\overline{\gamma}_{\mathrm{RGNN}}$ over a sparsity sweep, compare Figure 2 (a), so smaller area indicates a tighter bound. Panels vary **(a)** width $m$, **(b)** number of graphs $|\mathcal{D}_{\mathrm{syn}}|$, **(c)** number of nodes per graph $n$, **(d)** number of message-passing layers $L$, and **(e)** number of relations $|\mathcal{R}|$. For each $\rho$, $\gamma_{\mathrm{empirical}}$ is estimated from 10 pruning-mask trials, while $\gamma_{\mathrm{RGNN}}$ is instantiated using model-specific statistics and averaged over 10 sparse-mask samples. Unless varied per panel: $m = 16$, $|\mathcal{D}_{\mathrm{syn}}| = 30$, 6 nodes per graph, $L = 3$, and $|\mathcal{R}| = 3$.

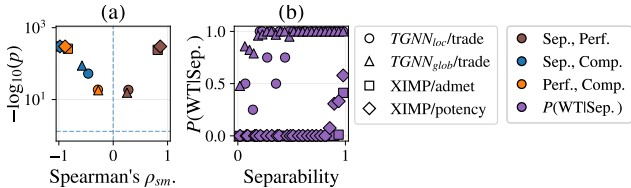

*Figure 5.* **(a)** For each model–dataset variant, we compute Spearman's $\rho_{sm}$ (and $p$-value) across runs between (i) separability (Sep.) and normalized test performance (Perf.), (ii) Sep. and compression (Compr.), and (iii) Perf. and Compr.; points plot $(\rho_{sm}, -\log_{10} p)$ (log-$y$), with colors encoding the variable pair and markers the model–dataset group; dashed lines indicate $\rho_{sm} = 0$ and $p = 0.05$. **(b)** $P(\mathrm{WT} \mid \mathrm{Sep.})$ for pruned runs, shown per model–dataset variant over separability bins (midpoints on $x$).

$\rho \in [0.65, 0.95]$), and tightens again as success vanishes for $\rho \to 1$. The observed variations of $\overline{\gamma}_{\mathrm{RGNN}}$ in Figure 2(a) reflect the dependency of $\gamma_{\mathrm{RGNN}}$ on statistics obtained from concrete model and mask instantiations. This matches the standard deviation (0.18) of the empirical fraction of distinguishable graphs over $\rho \in [0.65, 0.95]$. Figure 4 studies bound tightness through the integrating the gap $\Delta_\gamma$. The clearest effect is width: as $m$ increases, the gap decreases, indicating that once the model is sufficiently overparameterized, the lower bound tracks the empirical success curve more closely. Increasing $|\mathcal{D}_{\mathrm{syn}}|$ has the opposite effect and makes the guarantee more conservative, which is consistent with the need to control more witnessed inputs and distinctions on the finite $\mathcal{D}_{\mathrm{syn}}$. The trends with $n$ and $|\mathcal{R}|$ require more caution. In our setup, varying $n$ and $|\mathcal{R}|$ does not only change a single quantity in the bound; it also changes the synthetic graphs generated (see Appendix G), and, therefore, the witnessed statistics used to instantiate $\gamma_{\mathrm{RGNN}}$. Hence, a smaller gap in panels (c) and (e) should not be read as saying that larger graphs or more relations intrinsically improve the lower bound itself. Rather, these changes move the sparsity regimes in which both $\gamma_{\mathrm{emp.}}$ and $\overline{\gamma}_{\mathrm{RGNN}}$ curves are closer (compare Figure 2(a)). Thus, Figure 4 complements rather than contradicts Figure 1: Figure 1 analyzes how the bound varies as a function of its governing quantities, whereas Figure 4 measures how well the instantiated bound matches empirical behavior for a specific construction.

Figure 2(b) shows that higher pre-training separability is associated with stronger gradient flow, lower final training loss, and higher final training accuracy. Figure 3 strengthens this conclusion. Even at fixed sparsity, higher pre-training separability remains significantly associated with stronger gradient flow and lower final loss across all tested sparsity levels, and with higher accuracy at the higher sparsity levels. Thus, the optimization trend in Figure 2(b) is not merely a byproduct of larger subnetworks, but persists in a within-sparsity comparison as well. This matches Section 3's optimization discussion: masks that preserve more label-relevant distinctions at initialization retain more gradient-connected paths, so loss signals reach more surviving weights. Moreover, this aligns with the hypothesis that more expressive sparse RGNN initializations are easier to optimize, which is consistent with Kummer et al. (2025b)'s empirical findings and theoretical Gradient Diversity (Yin et al., 2018) analysis for classic GNNs. Figure 5 further supports this on real-world benchmarks and practical architectures: pre-training separability correlates with and predicts test-set performance and is anticorrelated with pruning compression. In practice, this highlights the importance of expressivity-aware sparsification for finding winning lottery tickets.

## 5. Conclusion

We introduce RSELTH, proving that sufficiently wide RGNNs contain sparse subnetworks that match 1-RWL expressivity on any finite dataset and deriving an explicit lower bound $\gamma_{\mathrm{RGNN}}$ on the probability that a random pruning mask yields such a subnetwork. We show the theory extends to TGNNs and XIMP/HIMP-type architectures. Our results indicate that (i) $\gamma_{\mathrm{RGNN}}$ is tight in the high- and low-success regimes and becomes most conservative near the critical sparsity threshold where expressivity breaks down, (ii) only a narrow band of width–sparsity configurations yields cost-efficient, highly expressive sparse initializations, (iii) the attainable expressivity at fixed width and sparsity depends strongly on dataset complexity and (iv) more expressive sparse initializations can be easier to optimize. Together, our results position RSELTH as a unifying perspective on sparse, expressive RGNNs and TGNNs and beyond.

## Acknowledgements

This work was supported in part by the Vienna Science and Technology Fund (WWTF) and the City of Vienna through grants [10.47379/VRG19009] and [10.47379/ICT22059].

## Impact Statement

This work advances the theoretical understanding of how parameter sparsity interacts with expressivity in relational and temporal graph neural networks. By clarifying when random pruning can preserve discriminative power, the results may support the design of more compute- and memory-efficient graph models, which can reduce deployment costs and energy use in applications where graph learning is already standard.

The methods studied here are general and could be applied to sensitive relational or temporal data (e.g., social, financial, or behavioral graphs). Improved efficiency can lower the barrier to deploying such models at scale, which may amplify risks around privacy, profiling, or biased decision-making if used without appropriate safeguards.

Our contributions do not introduce new data collection or new capabilities targeted at a specific domain, and they do not address fairness, privacy, or security directly. Practitioners should treat the theory as one component in a broader responsible pipeline, including dataset governance, privacy-preserving training where appropriate, and domain-specific evaluations for bias and misuse.

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

## A. TGNN → RGNN via refined relational alphabet

The construction of a $\widetilde{K}_\star(TG)$ using an augmented relation alphabet is a little more intricate. For a fixed temporal graph $TG = ((G_1, t_1), \ldots, (G_n, t_n))$ in $\mathcal{D}$ with strictly increasing timestamps, let the finite set of realized gaps as $\Delta(TG) := \{ t_j - t_i \mid 1 \le i \le j \le n, \text{and there is a corresponding edge in } K_\star(TG) \}$. Furthermore, let $\sigma : \Delta(TG) \to \{1, \ldots, B\}$ be any injective enumeration of these gaps (with $B := |\Delta(TG)|$). We define the *refined* relation alphabet as $\widetilde{\mathcal{R}} := \{ (r, b) \mid r \in \{0, \ldots, n-1\}, \ b \in \{1, \ldots, B\}, \text{and } \exists i \le j : \ j-i = r, \ \sigma(t_j - t_i) = b \}$. We then form a refined temporal knowledge graph $\widetilde{K}_\star(TG) = (V^{\text{time}}, \dot{E}^\star, \lambda^{\text{time}}, \tilde{\tau}^\star)$ by keeping the same node set and directed edge set as in $K_\star(TG)$, but labeling each edge $((v, t_i) \to (u, t_j))$ by $\tilde{\tau}^\star((v, t_i), (u, t_j)) := (j-i, \ \sigma(t_j - t_i)) \in \widetilde{\mathcal{R}}$. Then, we write the per-relation, per-direction neighborhoods on $\widetilde{K}_\star(TG)$ as $N^-_{(r,b)}(x) = \{ y \mid \tilde{\tau}^\star(x, y) = (r, b) \}$ and $N^+_{(r,b)}(x) = \{ y \mid \tilde{\tau}^\star(y, x) = (r, b) \}$.

**Lemma A.1.** *Let $\star \in \{\text{glob}, \text{loc}\}$ and a TGNN layer as in Eqs.* (4)-(5) *with maps $\Omega_a, \Omega_c, \Phi^{(l)}, \Xi, \zeta$. Consider the RGNN instantiated on the refined temporal knowledge graph $\widetilde{K}_\star(TG)$ with relation alphabet $\widetilde{\mathcal{R}}$. There exist per-relation message maps $\Phi^{(l)}_{(r,b),\pm} : \mathbb{R}^d \to \mathbb{R}^d$ and a combiner $\Gamma^{(l)}$ (same as in Eq.* (5)) *such that, for all $x \in V^{\text{time}}$,*

$$
m^{(l)}_{(r,b),-}(x) = \Omega_a\Big( \{\!\!\{ \Phi^{(l)}_{(r,b),-}(h_y^{(l)}) \mid y \in N^-_{(r,b)}(x) \}\!\!\} \Big),
$$
$$
m^{(l)}_{(r,b),+}(x) = \Omega_a\Big( \{\!\!\{ \Phi^{(l)}_{(r,b),+}(h_y^{(l)}) \mid y \in N^+_{(r,b)}(x) \}\!\!\} \Big),
$$
$$
h_x^{(l+1)} = \Gamma^{(l)}\Big( \Omega_c(\{\!\!\{ h_x^{(l)} \}\!\!\} \uplus \mathcal{M}^{(l)}(x)) \Big).
$$

*with $\mathcal{M}^{(l)}(x) := \biguplus_{s \in \widetilde{\mathcal{R}}} \{\!\!\{ m^{(l)}_{s,-}(x), \ m^{(l)}_{s,+}(x) \}\!\!\}$ and these updates* exactly *match those of the original TGNN on $TG$ (global or local, respectively).*

*Proof.* Let $x = (v, t_j) \in V^{\text{time}}$ and $\star \in \{\text{glob}, \text{loc}\}$. Let $\mathcal{N}_\star(x)$ denote the multiset of neighbors used by the TGNN aggregation at $x$ in Eq. (4): for $\star = \text{glob}$, $\mathcal{N}_\star(x) = \{(u, t_i) \mid \{u, v\} \in E_i, \ i \le j\}$; for $\star = \text{loc}$, $\mathcal{N}_\star(x) = \{(u, t_j) \mid \exists i \le j : \{u, v\} \in E_i\}$.

Each $y \in \mathcal{N}_\star(x)$ is associated with a unique *index lag* $r = j - i$ and a *realized time gap* $\Delta = t_j - t_i$. By construction of the augmented relation alphabet $\widetilde{\mathcal{R}}$, we assign a bucket $b = \sigma(\Delta)$ to $\Delta$. Hence every neighbor $y$ determines a unique triple $(r, b, \triangleright)$, whereby $\triangleright \in \{+, -\}$ indicates whether the corresponding directed edge in $\widetilde{K}_\star(TG)$ enters or leaves $x$.

This induces a disjoint partition of $\mathcal{N}_\star(x)$ into the per-relation/per-direction neighborhoods

$$
\mathcal{N}_\star(x) = \biguplus_{(r,b) \in \widetilde{\mathcal{R}}} \Big( N^+_{(r,b)}(x) \uplus N^-_{(r,b)}(x) \Big),
$$

where, by definition of $\widetilde{K}_\star(TG)$, $N^\pm_{(r,b)}(x) = \{ y \in V^{\text{time}} \mid (r, b) \text{ is the edge type and } y \xrightarrow{\pm} x \}$.

For each bucket $b$ fix the (unique) gap value realized on that bucket at $x$ and denote it by $\Delta_b$. Define the per-relation message maps

$$
\Phi^{(l)}_{(r,b),\pm}(h) := \Phi^{(l)}(\Xi(h, \zeta(\Delta_b))).
$$

Because every $y \in N^\pm_{(r,b)}(x)$ has the same index lag $r$ and the same realized gap $\Delta_b$, we obtain

$$
\Omega_a\Big( \{\!\!\{ \Phi^{(l)}_{(r,b),\pm}(h_y^{(l)}) \mid y \in N^\pm_{(r,b)}(x) \}\!\!\} \Big) = \Omega_a\Big( \{\!\!\{ \Phi^{(l)}(\Xi(h_y^{(l)}, \zeta(\Delta_b))) \mid y \in N^\pm_{(r,b)}(x) \}\!\!\} \Big)
$$

for parameterizations of $\Phi^{(l)}_{(r,b),\pm}$ suitable to the chosen of $\Xi, \zeta$ (e.g., $\|, \Delta_b$, respectively, consistent with Wałęga & Rawson (2025).

Given $\zeta(\Delta_b) = \zeta(t_j - t_i)$ for all $y \in N^\pm_{(r,b)}(x)$, the right-hand side is exactly the contribution that the original TGNN aggregation, Eq. (4), would collect from the subset $N^\pm_{(r,b)}(x)$ of neighbors.

Since $\{N^+_{(r,b)}(x), N^-_{(r,b)}(x)\}_{(r,b)}$ forms a disjoint partition of the TGNN's neighborhood $\mathcal{N}_\star(x)$, aggregating *within* each block by $\Omega_a$ and then feeding the family of block summaries to the combiner reproduces the TGNN's total message: the multiset

$$\{\!\{\Phi^{(l)}(\Xi(h_y^{(l)}, \zeta(t_j - t_i))) \mid y \in \mathcal{N}_\star(x)\}\!\}$$

is the disjoint multiset union of the blockwise multisets $\{\!\{\Phi^{(l)}_{(r,b),\pm}(h_y^{(l)}) \mid y \in N^\pm_{(r,b)}(x)\}\!\}$. Therefore, with the same $\Omega_c$ and $\Gamma^{(l)}$ as in (5), the update

$$h_x^{(l+1)} = \Gamma^{(l)}\Big( \Omega_c(\{\!\{h_x^{(l)}\}\!\} \uplus \biguplus_{(r,b)} \{\!\{m^{(l)}_{(r,b),-}(x), m^{(l)}_{(r,b),+}(x)\}\!\}) \Big)$$

coincides with applying $\Gamma^{(l)}$ to the combine of $h_x^{(l)}$ and the TGNN's one-shot aggregate $m^{(l)}_{t_j}(v)$ from (4). Hence the relational layer above reproduces *exactly* the original TGNN layer on $TG$ (for both glob and loc). $\qquad\square$

While correct and allowing for the unifying perspective through RSELTH similar as Lemma 3.3, as Theorem 3.1 applies verbatim to the above RGNN, this reduction has a drawback: the resulting sparsity (pruning) masks for the RGNN do not, in general, trivially transfer back to the original TGNN, as handling the $\delta_{ij}$ present in Eqs. (4)-(5) in the RGNN framework given by Eqs. (1)-(3) requires a refinement of the relation alphabet along with an associated extended parameterization.

## B. Temporal SELTH (TSELTH)

While Temporal SELTH as stated below in Theorem B.1 directly follows from the RGNN reformulation discussed in the main paper, we provide a reformulation and standalone proof for completeness.

**Theorem B.1** (Temporal SELTH). *Let $\mathcal{D}$ be a finite collection of finite temporal graphs. Let $\Psi^{(L)}$ be a corresponding sufficiently overparameterized depth-$L$ TGNN of the pattern described either in Eqs. (4)-(5).Then, for $M \sim^{i.i.d.} \mathscr{B}(1 - \rho)$, $\Theta_0 \sim \mathscr{U}_c^d$ and corresponding sparse initialization $\widehat{\Theta}_0 = M \odot \Theta_0$ of $\Psi^{(L)}$, the following holds for all $TG_a, TG_b \in \mathcal{D}$,*

$$K_\star(TG_a) \not\simeq_{\mathrm{RWL}^{(L)}} K_\star(TG_b) \iff \tag{9}$$

$$\widehat{\Psi}^{(L)}_{\widehat{\Theta}_0,\star}(TG_a) \neq \widehat{\Psi}^{(L)}_{\widehat{\Theta}_0,\star}(TG_b) \tag{10}$$

*with a probability at least $\gamma_{\mathrm{TGNN}} > 0$.*

*Proof.* For any local or global TGNN, Lemma A.1 implies that, at layer $l$, the set of inputs that the message MLP $\Phi^{(l)}$ actually receives across $\mathcal{D}$ is given by

$$S^{(l)} := \bigcup_{TG \in \mathcal{D}} \bigcup_{x \in V^{\mathrm{time}}(TG)} \bigcup_{(r,b,\rhd)} \left\{ \Xi(h_y^{(l)}, \zeta(\Delta(y,x))) \mid y \in N^\rhd_{(r,b)}(x) \right\},$$

where $(r,b,\rhd)$ ranges over the per-lag, per-bucket, per-direction neighborhoods $N^\rhd_{(r,b)}(\cdot)$ on $\tilde{K}_\star(TG)$ (see Lemma A.1), and $\Delta(y,x)$ is the time gap used by the TGNN.

On a finite $\mathcal{D}$ over finite temporal graphs, $N_l := |S^{(l)}|$ is likewise finite. Let the minimum $\ell_0$-separation across distinct inputs be

$$s_l := \min_{\substack{z \neq z' \\ z,z' \in S^{(l)}}} \|z - z'\|_0 \quad (\geq 1).$$

Consider the MLP with width $m_l$ and a Bernoulli pruning mask whose entries are 0 independently with probability $\rho \in (0,1)$ (sparsity ratio). By Lemma A.1 in Kummer et al. (2025b), the masked map is injective on $S_l$ with probability at least

$$\gamma_l \geq 1 - \binom{N_l}{2} \rho^{s_l m_l}.$$

and the layer is injective with probability

$$\gamma_l^{\mathrm{layer}} \geq 1 - \binom{N_l}{2} \rho^{s_l m_l} - \binom{N_{l,\mathrm{comb}}}{2} \rho^{s_{l,\mathrm{comb}} m_{l,\mathrm{comb}}},$$

where $(\cdot)_{\text{comb}}$ refers to the combine MLP. As in Theorem 3.1, we define for the combine MLP the input multiset $M_l$ (which here is similarly the finite multiset of possible branch summaries), width $m_{l,\text{comb}}$, $N_{l,\text{comb}} = |M_l|$ and

$$s_{l,\text{comb}} := \min_{\substack{z \neq z' \\ z,z' \in M_l}} \|z - z'\|_0 \quad (\geq 1).$$

If we use independent masks across all MLP blocks, and take a conservative worst-case bound with

$$N_{\max} := \max_l N_l, \quad s_{\min} := \min_l s_l, \quad m_{\min} := \min_l m_l,$$

$$N_{\max,\text{comb}} := \max_l N_{l,comb}, \quad s_{\min,} := \min_l s_{l,comb}, \quad m_{\min,\text{comb}} := \min_l m_{l,comb},$$

then each branch is injective with probability at least $1 - \binom{N_{max}}{2}\rho^{s_{\min}m_{\min}}$ and each combination with probability at least $1 - \binom{N_{max,\text{comb}}}{2}\rho^{s_{\min,\text{comb}}m_{\min,\text{comb}}}$. For $M = 1$ MLP sublayers per MP layer and $L$ MP layers, without relational branch MLPs as in Theorem 3.1, the crude product lower bound for TGNNs simplifies to

$$\gamma_{\text{TGNN}} \geq \left( \left[\left(1 - \binom{N_{max}}{2}\right)\rho^{s_{\min}m_{\min}}\right]_+ \right)\left(\left[1 - \binom{N_{\max,\text{comb}}}{2}\right)\rho^{s_{\min,\text{comb}}m_{\min,\text{comb}}}\right]_+\right) \right)^L$$

with $[x]_+ := \max\{x, 0\}$ and furthermore to

$$\widetilde{\gamma}_{\text{TGNN}} \geq \left( \left[1 - \binom{\widetilde{N}_{max}}{2}\right)\rho^{\widetilde{s}_{\min}\widetilde{m}_{\min}}\right]_+\right)^L,$$

with

$$\widetilde{N}_{\max} := \max\{N_{\max}, N_{\max,\text{comb}}\}, \quad \widetilde{s}_{\min} := \min\{s_{\min}, s_{\min,comb}\}, \quad \widetilde{m}_{\min} := \min\{m_{\min}, m_{\min,\text{comb}}\}.$$

By the same argument as in Theorem 3.1, we argue this estimate is conservative; so for fixed $N_{max}, s_{\min}$ and their combine or $\widetilde{\cdot}$ counterparts, one can pick $m$ such that $\gamma_{\text{TGNN}} > 0$. $\qquad\square$

## C. TSELTH Node Level Transfer

The transfer is possible because 1-RWL (and its temporal variants via $K_\star(TG)$) operates fundamentally through iterative node color refinement: nodes with distinct $k$-hop relational or temporal neighborhoods receive distinct colors (Corollary 2.2), encoding their structural roles. In sparse subnetworks under SELTH, the preservation of injective aggregation and combination functions ensures that node embeddings $h_x^{(L)}$ remain distinct for such nodes, providing the discriminative features needed for node-level predictions.

**Lemma C.1.** *Fix $\star \in \{\text{glob}, \text{loc}\}$ and depth $L$. Consider a TGNN of the form Eqs. (4)-(5) with (i) an injective label encoder, (ii) injective $\Phi^{(l)}$ and $\Gamma^{(l)}$ on the* finite *input sets they actually witness on a finite dataset $\mathcal{D}$, and (iii) $\Omega_a, \Omega_c$ chosen as injective multiset encoders on those finite domains (e.g., $\sum$ with a linearly independent codebook (see Remark 2.1)). Then for every $TG \in \mathcal{D}$ and every $k \leq L$ there exists an injective $\eta_k$ such that for all timestamped nodes $x$*

$$h_x^{(k)} = \eta_k(c_{K_\star(TG)}^{(k)}(x)).$$

*In particular, if $c_{K_\star(TG)}^{(k)}(x) \neq c_{K_\star(TG)}^{(k)}(x')$, then $h_x^{(k)} \neq h_{x'}^{(k)}$ under the same parameters.*

*Proof.* We proceed by induction on $k$.

*Base case ($k = 0$).* By definition of directed 1-RWL on $K_\star(TG)$, the initial color is $c_{K_\star(TG)}^{(0)}(x) = \lambda^{\text{time}}(x)$. The TGNN initializes $h_x^{(0)} = \Lambda(\lambda^{\text{time}}(x))$ with an injective label encoder $\Lambda$. Hence setting $\eta_0 := \Lambda$ gives $h_x^{(0)} = \eta_0(c_{K_\star(TG)}^{(0)}(x))$, and $\eta_0$ is injective.

*Inductive step.* Assume for some $k \geq 0$ there exists an injective $\eta_k$ such that $h_y^{(k)} = \eta_k(c_{K_\star(TG)}^{(k)}(y))$ for all timestamped nodes $y$. Fix $x = (v, t_j) \in V^{\text{time}}$. In the global/local TGNN, aggregation ranges over the temporal neighborhood $N(v, t_j)$

as in Eq. (4). By construction of $K_\star(TG)$, each $(u, t_i) \in N(v, t_j)$ with $i \leq j$ appears as an incoming edge $(u, t_i) \to (v, t_j)$ labeled by the unique index lag $r = j - i$. Writing

$$N_r^+(x) := \{ (u, t_i) \mid (u, t_i) \to (v, t_j) \text{ in } K_\star(TG), \ j - i = r \},$$

we obtain a *disjoint partition* of the temporal neighborhood by lag:

$$N(v, t_j) = \biguplus_{r=0}^{j-1} N_r^+(x),$$

and, moreover, $\delta((u, t_i), x) = t_j - t_i$ is constant on each slice $N_r^+(x)$ (namely $t_j - t_i$ with $j - i = r$). Consequently, we may equivalently index the aggregation by the per-lag, per-direction neighborhoods $N_r^\triangleright(x)$ of $K_\star(TG)$, which matches the directed 1-RWL bookkeeping; the same reasoning applies to the outgoing branch $\triangleright = -$ if present.

For each relation $r$ and direction $\triangleright \in \{+, -\}$ on $K_\star(TG)$, define

$$\mathcal{A}_{r,\triangleright}^{(k)}(x) := \{\!\!\{\, c_{K_\star(TG)}^{(k)}(y) \mid y \in N_r^\triangleright(x) \,\}\!\!\} \quad \text{and} \quad \mathcal{H}_{r,\triangleright}^{(k)}(x) := \{\!\!\{\, h_y^{(k)} \mid y \in N_r^\triangleright(x) \,\}\!\!\}.$$

By the induction hypothesis, elementwise $h_y^{(k)} = \eta_k(c_{K_\star(TG)}^{(k)}(y))$ on the *finite* dataset $\mathcal{D}$, so (as multisets)

$$\mathcal{H}_{r,\triangleright}^{(k)}(x) = \eta_k[\mathcal{A}_{r,\triangleright}^{(k)}(x)],$$

where $\eta_k[\cdot]$ acts elementwise.

Apply the layer's elementwise message map and relation/direction aggregator:

$$m_{r,\triangleright}^{(k)}(x) = \Omega_a(\{\!\!\{\, \Phi^{(k)}(h_y^{(k)}) \mid y \in N_r^\triangleright(x) \,\}\!\!\}) = \underbrace{\Omega_a(\{\!\!\{\, \Phi^{(k)}(\eta_k(a)) \mid a \in \mathcal{A}_{r,\triangleright}^{(k)}(x) \,\}\!\!\})}_{:= F_{r,\triangleright}^{(k)}(\mathcal{A}_{r,\triangleright}^{(k)}(x))}.$$

By assumption, $\Phi^{(k)}$ is injective on the finite set $\eta_k(\mathrm{Im}\, c^{(k)})$ actually witnessed on $\mathcal{D}$, and $\Omega_a$ is an injective multiset encoder on the corresponding finite domain (e.g., sum over a linearly independent codebook; cf. Remark 2.1). Therefore $F_{r,\triangleright}^{(k)}$ is injective on the finite family of multisets $\mathcal{A}_{r,\triangleright}^{(k)}(x)$ realized on $\mathcal{D}$.

Next, combine the self-embedding and all per-$(r, \triangleright)$ messages:

$$h_x^{(k+1)} = \Gamma^{(k)}\Big(\Omega_c(\{\!\!\{\, h_x^{(k)} \,\}\!\!\} \uplus \biguplus_{r,\triangleright} \{\!\!\{\, m_{r,\triangleright}^{(k)}(x) \,\}\!\!\})\Big) = \Gamma^{(k)}\Big(\Omega_c(\{\!\!\{\, \eta_k(c^{(k)}(x)) \,\}\!\!\} \uplus \biguplus_{r,\triangleright} \{\!\!\{\, F_{r,\triangleright}^{(k)}(\mathcal{A}_{r,\triangleright}^{(k)}(x)) \,\}\!\!\})\Big).$$

By assumption, $\Omega_c$ is injective on the finite domain of inputs realized on $\mathcal{D}$, and $\Gamma^{(k)}$ is injective there as well. Moreover, since each $F_{r,\triangleright}^{(k)}$ is injective and we retain the $(r, \triangleright)$ tags, the map from the refinement input

$$\Big(c^{(k)}(x), \ (\mathcal{A}_{r,-}^{(k)}(x))_r, \ (\mathcal{A}_{r,+}^{(k)}(x))_r\Big)$$

to all per-relation/per-direction messages $m_{r,\triangleright}^{(k)}(x)$ (for $r \in \mathcal{R}$ and $\triangleright \in \{+, -\}$) is injective. Applying the injective $\Omega_c$ to this indexed tuple and then the injective $\Gamma^{(k)}$ preserves injectivity, so the right-hand side is an *injective function* of the above refinement input, which is precisely the directed 1-RWL refinement input at $x$ on $K_\star(TG)$. As the 1-RWL color update

$$c_{K_\star(TG)}^{(k+1)}(x) = \Upsilon_3\Big(c^{(k)}(x), \ (\mathcal{A}_{r,-}^{(k)}(x))_r, \ (\mathcal{A}_{r,+}^{(k)}(x))_r\Big)$$

uses an injective $\Upsilon_3$ on this finite domain, we can define $\eta_{k+1}$ on the *set of $(k+1)$-colors actually realized on $\mathcal{D}$* by

$$\eta_{k+1}(c_{K_\star(TG)}^{(k+1)}(x)) := \Gamma^{(k)}\Big(\Omega_c(\{\!\!\{\, \eta_k(c^{(k)}(x)) \,\}\!\!\} \uplus \biguplus_{r,\triangleright} \{\!\!\{\, F_{r,\triangleright}^{(k)}(\mathcal{A}_{r,\triangleright}^{(k)}(x)) \,\}\!\!\})\Big).$$

This is well-defined (equal colors correspond to equal refinement inputs) and injective (distinct colors correspond to distinct refinement inputs and the outer map is injective), and we have $h_x^{(k+1)} = \eta_{k+1}(c_{K_\star(TG)}^{(k+1)}(x))$.

By induction, for every $k \leq L$ there exists an injective $\eta_k$ with $h_x^{(k)} = \eta_k(c_{K_\star(TG)}^{(k)}(x))$ for all timestamped nodes $x$. In particular, for any $x, x'$, if $c_{K_\star(TG)}^{(k)}(x) \neq c_{K_\star(TG)}^{(k)}(x')$, then $h_x^{(k)} \neq h_{x'}^{(k)}$ under the same parameters. $\qquad\square$

By Theorem B.1, on any finite $\mathcal{D}$ there exists a *sparse* mask $M$ that preserves the layerwise injectivity conditions above. Therefore, Lemma C.1 continues to hold under the same sparse support $M$.

## D. Proofs (Main Paper)

Complete proofs for all theoretical claims made in the main paper are given below. The proofs for theoretical claims made in the appendix are given directly below the respective claim.

### D.1. Proof of Corollary 2.2

Corollary 2.2 is effectively a concise summary of Theorems 2 to 6 proven by Wałęga & Rawson (2025).

*Proof.* Let a temporal graph $TG$ and timestamped nodes $x = (v, t_i), x' = (u, t_j) \in V^{\text{time}}$, and let $\star \in \{\text{glob}, \text{loc}\}$ match the TGNN MP type.

($\Rightarrow$) By Theorem 2 (global) and Theorem 5 (local) of Wałęga & Rawson (2025), directed 1-RWL on $K_\star(TG)$ upper-bounds any MP-TGNN of the corresponding type. In our notation this reads: if $c^{(k)}_{K_\star(TG)}(x) = c^{(k)}_{K_\star(TG)}(x')$, then there exists a TGNN layer of the form (4) with combiner (5) for which we have $h^{(k)}_x = h^{(k)}_{x'}$, which implies the existence of corresponding parameters.

($\Leftarrow$) By Theorem 3 (global) and Theorem 6 (local) of Wałęga & Rawson (2025), there exists a TGNN of the corresponding architectural form such that, layerwise, embeddings can be mapped 1-to-1 to the 1-RWL colors on $K_\star(TG)$. Instantiating our layer class (4), (5) with $\Omega_a = \sum$ and choosing $\Phi^{(l)}, \Gamma^{(l)}$ as in the referred constructions, hence implies the existence of parameters for which

$$h^{(k)}_x = h^{(k)}_{x'} \quad \Longleftrightarrow \quad c^{(k)}_{K_\star(TG)}(x) = c^{(k)}_{K_\star(TG)}(x').$$

This proves the bi-implication for some parametrization of any TGNN of the examined class. $\qquad\square$

### D.2. Proof of Lemma 3.2

*Proof.* Let $\star \in \{\text{glob}, \text{loc}\}$ and $TG_a, TG_b \in \mathcal{D}$ and assume $K_\star(TG_a) \not\simeq_{\text{RWL}^{(L)}} K_\star(TG_b)$. Then their $L$-step 1-RWL color multisets on $K_\star$ differ:

$$\{\!\!\{\, c^{(L)}_{K_\star(TG_a)}(x) \mid x \in V^{\text{time}}(TG_a) \,\}\!\!\} \;\neq\; \{\!\!\{\, c^{(L)}_{K_\star(TG_b)}(x) \mid x \in V^{\text{time}}(TG_b) \,\}\!\!\}.$$

By Corollary 2.2, there exists a parametrization of the (local/global) TGNN such that, layerwise, $h^{(L)}_x = \psi_L(c^{(L)}_{K_\star(TG)}(x))$ for some injective $\psi_L$. Consequently, the two multisets of node embeddings differ.

Let the TGNN's graph readout $\chi$ be permutation-invariant and injective on finite multisets; for instance, $\chi(\{\!\{h_x\}\!\}) = \alpha(\sum_x h_x)$, where $\alpha$ is injective (e.g., a suitable MLP (Amir et al., 2024; Puthawala et al., 2022)) and the codebook (see Remark 2.1) $\{\psi_L(\cdot)\}$ is chosen linearly independent, which makes the sum an injective multiset encoder (Zaheer et al., 2017). Then

$$\Psi^{(L)}_{\Theta,\star}(TG_a) \;=\; \chi(\{\!\{h^{(L)}_x \mid x \in V^{\text{time}}(TG_a)\}\!\}) \;\neq\; \chi(\{\!\{h^{(L)}_x \mid x \in V^{\text{time}}(TG_b)\}\!\}) \;=\; \Psi^{(L)}_{\Theta,\star}(TG_b),$$

for suitable $\Theta$ as claimed.

Conversely, assume $\Psi^{(L)}_{\Theta,\star}(TG_a) \neq \Psi^{(L)}_{\Theta,\star}(TG_b)$. By injectivity of $\chi$, the underlying multisets of node embeddings must differ:

$$\{\!\{h^{(L)}_x \mid x \in V^{\text{time}}(TG_a)\}\!\} \;\neq\; \{\!\{h^{(L)}_x \mid x \in V^{\text{time}}(TG_b)\}\!\}.$$

As $h^{(L)}_x = \psi_L(c^{(L)}_{K_\star(TG)}(x))$ with injective $\psi_L$, the two color multisets at depth $L$ must differ, hence $K_\star(TG_a) \not\simeq_{\text{RWL}^{(L)}} K_\star(TG_b)$.

This establishes the claimed bi-implication for the chosen parameterization $\Theta$. $\qquad\square$

### D.3. Proof of Theorem 3.1

*Proof.* In R-GNN $\Psi^{(L)}$, if each layer injectively aggregates separately per relation and per direction (Eq. (1)) and then applies an injective combine to the resulting tuple together with the node's own embedding (Eq. (3)), the layer realizes exactly the 1-RWL refinement step at the next iteration. This is the relational analogue of the criterion used in the static case (Kummer et al., 2025b), Lemma 2.1, which is based on the correspondence between 1-WL and GNN aggregation/combine injectivity (Xu et al., 2019). Stacking $L$ such RGNN layers matches 1-RWL to depth $L$.

Hence, we need to first show injectivity can be preserved under pruning. Let a finite dataset $\mathcal{D}$ (cardinality $|\mathcal{D}|$) and a depth-$L$ RGNN $\Psi^{(L)}$. At any layer $l$ and for any *branch* $b$ (one per relation $r \in \mathcal{R}$ in the undirected case, or per relation/direction in the directed case), let $X_{l,b} \subset \mathbb{R}^{n_{l,b}}$ denote the finite set of inputs that actually occur at branch $b$ when propagating $\mathcal{D}$ up to layer $l$. We write $N_{l,b} = |X_{l,b}|$ and let

$$s_{l,b} := \min_{\substack{x \neq x' \\ x,x' \in X_{l,b}}} \|x - x'\|_0 \quad (\geq 1),$$

be the minimum $\ell_0$-separation across distinct inputs in $X_{l,b}$. Consider the branch MLP $\Phi_b^{(l)}$ with width $m_{l,b}$ and a Bernoulli pruning mask whose entries are 0 independently with probability $\rho \in (0,1)$ (sparsity ratio). By Lemma A.1 in Kummer et al. (2025b), which has distributional assumptions on parameters and pruning masks, the masked branch map is injective on $X_{l,b}$ with probability at least

$$\gamma_{l,b} \geq 1 - \binom{N_{l,b}}{2} \rho^{s_{l,b}\, m_{l,b}}.$$

Apply the same argument to every branch of layer $l$ and to the combine MLP $\Gamma^{(l)}$ (whose input multiset $M_l$ is the finite multiset of possible branch summaries) with width $m_{l,\mathrm{comb}}$, $N_{l,\mathrm{comb}} = |M_l|$ and

$$s_{l,\mathrm{comb}} := \min_{\substack{x \neq x' \\ x,x' \in M_l}} \|x - x'\|_0 \quad (\geq 1).$$

Then, by a union bound one obtains a layer-level lower bound

$$\gamma_l^{\mathrm{layer}} \geq 1 - \sum_{b \in \mathcal{B}_t} \binom{N_{l,b}}{2} \rho^{s_{l,b}\, m_{l,b}} - \binom{N_{l,\mathrm{comb}}}{2} \rho^{s_{l,\mathrm{comb}}\, m_{l,\mathrm{comb}}},$$

where $\mathcal{B}_l$ is the set of branches at layer $l$ and $(\cdot)_{\mathrm{comb}}$ refers to the combine MLP.

If we use independent masks across all MLP blocks, and take a conservative worst-case bound with

$$N_{\max} := \max_{l,b} N_{l,b}, \quad s_{\min} := \min_{l,b} s_{l,b}, \quad m_{\min} := \min_{l,b} m_{l,b}$$

$$N_{\max,\mathrm{comb}} := \max_l N_{l,comb}, \quad s_{\min,} := \min_l s_{l,comb}, \quad m_{\min,\mathrm{comb}} := \min_l m_{l,\mathrm{comb}},$$

then each branch is injective with probability at least $1 - \binom{N_{max}}{2} \rho^{s_{\min} m_{\min}}$ and each combination with probability at least $1 - \binom{N_{max,\mathrm{comb}}}{2} \rho^{s_{\min,\mathrm{comb}} m_{\min,\mathrm{comb}}}$. Counting $M$ MLP sublayers per MP layer and $L$ MP layers, with $|\mathcal{B}_l| = |\mathcal{R}|$ (undirected) or $|\mathcal{B}_l| = 2|\mathcal{R}|$ (directed), a crude product lower bound is

$$\gamma_{\mathrm{RGNN}} \geq \left( \left[ 1 - \binom{N_{\max}}{2} \rho^{s_{\min} m_{\min}} \right]_+ \right)^{|\mathcal{B}| M} \left( \left[ 1 - \binom{N_{max,\mathrm{comb}}}{2} \rho^{s_{\min,\mathrm{comb}} m_{\min,\mathrm{comb}}} \right]_+ \right)^L,$$

with $[x]_+ := \max\{x, 0\}$, where $|\mathcal{B}| = |\mathcal{R}|$ or $2|\mathcal{R}|$, which further simplifies to

$$\widetilde{\gamma}_{\mathrm{RGNN}} \geq \left( \left[ 1 - \binom{\widetilde{N}_{max}}{2} \rho^{\widetilde{s}_{\min} \widetilde{m}_{\min}} \right]_+ \right)^{L\,(M\,|\mathcal{B}|+1)},$$

with

$$\widetilde{N}_{\max} := \max\{N_{\max}, N_{\max,\mathrm{comb}}\}, \quad \widetilde{s}_{\min} := \min\{s_{\min}, s_{\min,comb}\}, \quad \widetilde{m}_{\min} := \min\{m_{\min}, m_{\min,\mathrm{comb}}\}.$$

This is conservative, as (i) earlier non-injectivity would shrink the finite input sets faced by later blocks, increasing their success probability and (ii) any increase of width $m$ drives $\rho^{sm}$ down exponentially, so for fixed $N_{max}, s_{\min}$ and their combine or $\widetilde{\cdot}$ counterparts, one can pick the width $m$ of the pruned branch and combine MLPs such that $\gamma_{\text{RGNN}} > 0$ or $\widetilde{\gamma}_{\text{RGNN}} > 0$.

Consequentially, the random pruning of $\Psi^{(L)}$ can preserve 1-RWL expressivity under mild assumptions[2].

$\square$

### D.4. Proof of Lemma 3.4

*Proof.* We begin by first showing the following auxiliary Corollary:

**Corollary D.1.** *Let $f : V(G_i^a) \to V(G_i^b)$ be the timewise isomorphism's snapshot-wise bijection and define $f' : (v, t_i^a) \mapsto (f(v), t_i^b)$. Then for all $l \in \{0, \ldots, L\}$ and all $x = (v, t_i^a)$ we have $c_{K_\star(TG_a)}^{(l)}(x) = c_{K_\star(TG_b)}^{(l)}(f'(x))$.*

*Proof.* By Wałęga & Rawson (2025)'s Theorem 10, timewise-isomorphic timestamped nodes produce *identical* embeddings in *any* TGNN at every layer. Instantiate the specific local/global TGNN that (by the standard WL-characterisation used in our Corollary 2.2) computes embeddings $h^{(l)} = \psi_l(c_{K_\star(\cdot)}^{(l)})$ with $\psi_l$ injective at each layer $l$. Applying Wałęga & Rawson (2025)'s Theorem 10 to this TGNN yields $h_x^{(l)} = h_{f'(x)}^{(l)}$, hence injectivity of $\psi_l$ gives $c_{K_\star(TG_a)}^{(l)}(x) = c_{K_\star(TG_b)}^{(l)}(f'(x))$ for all $l$. $\square$

The map $f'$ is a bijection between $V^{\text{time}}(TG_a)$ and $V^{\text{time}}(TG_b)$, so Corollary D.1 implies equality of the entire $L$-step color multisets, as stated. In particular, if one employs a permutation-invariant, multiset-injective graph readout $\chi$ for the final TGNN layer (e.g., a sum followed by an injective MLP), then applying $\chi$ to equal multisets yields equal graph-level outputs $\Psi_{\Theta,\star}^{(L)}(TG_a) = \Psi_{\Theta,\star}^{(L)}(TG_b)$ for suitable $\Theta$. $\square$

### D.5. Proof of Corollary 3.5

*Proof.* For each $(G, \{T_i, S_i\}) \in \mathcal{D}$, we build $G^\oplus$ as in Section 3. Then, by construction, stacking $L$ relational layers on $G^\oplus$ (with the appropriate parameter choices) simulates $L$ XIMP layers. As in the relational setup, we pick parameters so that at every layer $l$ there exists an injective map $\psi_l$ with $\widetilde{h}^{(l)} = \psi_l(c^{(l)})$, where $c^{(l)}$ is the $l$-round 1-RWL color on $G^\oplus$; this is feasible on finite inputs using sum aggregation with a linearly independent codebook (see Remark 2.1) and injective $\Phi_{r,+}^{(l)}, \Phi_{r,-}^{(l)}, \Gamma^{(l)}$.

If $G_a^\oplus \not\simeq_{\text{RWL}^{(L)}} G_b^\oplus$, then their $L$-round color multisets differ. Injectivity of $\psi_L$ implies the final-layer node-embedding multisets differ. With any permutation-invariant readout $\chi$ that is injective on finite multisets (e.g., sum followed by an injective MLP), the graph-level outputs differ:

$$\Psi_{\Theta}^{(L)}(G_a, \{T_i^a, S_i^a\}) \;\neq\; \Psi_{\Theta}^{(L)}(G_b, \{T_i^b, S_i^b\}).$$

Conversely, if $\Psi_{\Theta}^{(L)}(G_a, \{T_i^a, S_i^a\}) \neq \Psi_{\Theta}^{(L)}(G_b, \{T_i^b, S_i^b\})$ under an injective $\chi$, then the final-layer node-embedding multisets must differ; by injectivity of $\psi_L$, the $L$-round 1-RWL color multisets differ, hence $G_a^\oplus \not\simeq_{\text{RWL}^{(L)}} G_b^\oplus$.

Therefore, for suitable parameters, 1-RWL on $G^\oplus$ exactly characterizes the $L$-round distinguishing power of XIMP/HIMP on the finite collection $\mathcal{D}$. $\square$

---

[2]The non-colinearity result of Kummer et al. (2025b), Proposition 3.4, extends branch-wise to the relational block: with sufficiently wide layers and independently pruned weights, distinct inputs that are separated by the masked map almost surely produce non-colinear embeddings in each branch and after the combine, hence also at the graph readout.

### D.6. Proof of Lemma 3.3

*Proof.* Fix $\star \in \{\text{glob}, \text{loc}\}$ and a temporal node $x = (v, t_j) \in V^{\text{time}}$. Recall the TGNN aggregation and update from Eqs. (4)-(5):

$$\text{(global)} \quad m_{t_j}^{(l)}(v) = \Omega_a\Big(\{\!\!\{\Phi^{(l)}(\Xi(h_{(u,t_i)}^{(l)}, \zeta(\delta_{ij}))) \mid (u, t_i) \in N(v, t_j)\}\!\!\}\Big),$$

$$\text{(local)} \quad m_{t_j}^{(l)}(v) = \Omega_a\Big(\{\!\!\{\Phi^{(l)}(\Xi(h_{(u,t_j)}^{(l)}, \zeta(\delta_{ij}))) \mid (u, t_i) \in N(v, t_j)\}\!\!\}\Big),$$

$$h_{(v,t_j)}^{(l+1)} = \Gamma^{(l)}\Big(\Omega_c(\{\!\!\{h_{(v,t_j)}^{(l)}\}\!\!\} \uplus \{\!\!\{m_{t_j}^{(l)}(v)\}\!\!\})\Big),$$

where $\delta_{ij} = t_j - t_i$ and $N(v, t_j)$ is the temporal neighborhood $N(v, t_j) = \{(u, t_i) \mid \{u, v\} \in E_i,\ i \leq j\}$.

On the knowledge-graph side, let $K_\star(TG)$ be the (directed, typed) temporal knowledge graph constructed for $\star$ as in the preliminaries, and define its edge-feature augmentation $\xi_{(y \to x)} := \zeta(\delta(y, x))$, where for $y = (u, t_i)$ and $x = (v, t_j)$ we have $\delta(y, x) = t_j - t_i$. As in the statement, instantiate the RGNN on $\widetilde{K}_\star(TG)$ with

$$\widetilde{\Phi}^{(l)}(h, \xi) = \Phi^{(l)}(\Xi(h, \xi)),$$

strict tying $\Phi_{r,\pm}^{(l)} \equiv \Phi^{(l)}$ across $(r, \pm)$, and the *same* $(\Omega_a, \Omega_c, \Gamma^{(l)})$.

We now prove by induction on $l$ that for all $x = (v, t_j)$,

$$h_{(v,t_j)}^{(l)}\ \text{(TGNN)} \ = \ \widetilde{h}_{(v,t_j)}^{(l)}\ \text{(RGNN)}.$$

*Base case ($l = 0$).* Both models use the same initialization $h_{(v,t_j)}^{(0)} = \Lambda(\lambda^{\text{time}}(v, t_j))$, so the claim holds.

*Induction step.* Assume $h_{(\cdot)}^{(l)} = \widetilde{h}_{(\cdot)}^{(l)}$ for all timestamped nodes. Fix $x = (v, t_j)$ and consider the multiset of *incoming* neighbors to $x$ in $K_\star(TG)$, grouped by index lag $r = j - i$:

$$\bigcup_r N_r^+(x) \quad \text{with} \quad N_r^+(x) = \{(u, t_i) \mid (u, t_i) \to (v, t_j)\ \text{in}\ K_\star(TG),\ j - i = r\}.$$

By construction of $K_\star(TG)$, for every $(u, t_i) \in N(v, t_j)$ there is a unique index lag $r = j - i \in \{0, \ldots, n-1\}$ and a corresponding incoming edge of type $r$ into $x = (v, t_j)$: for $\star = \text{glob}$ the edge is $(u, t_i) \to (v, t_j)$, and for $\star = \text{loc}$ the edge is $(u, t_j) \to (v, t_j)$ but it is labeled by the witness snapshot $i$ via $r = j - i$. Hence each temporal neighbor $(u, t_i)$ belongs to exactly one lag class $r$, and the per-lag incoming neighborhoods form a disjoint (multi)set partition of the TGNN temporal neighborhood:

$$N(v, t_j) \ = \ \biguplus_r N_r^+(x).$$

Moreover, if $(u, t_i) \in N_r^+(x)$ is the (unique) witness for lag $r = j - i$, then the realized time gap used by the TGNN is fixed on that block:

$$\delta((u, t_i), x) \ = \ t_j - t_i \ = \ \delta_{ij}.$$

Define the RGNN's per-lag incoming messages as

$$m_{r,+}^{(l)}(x) \ = \ \Omega_a\Big(\{\!\!\{\widetilde{\Phi}^{(l)}(\widetilde{h}_y^{(l)}, \xi_{(y \to x)}) \mid y \in N_r^+(x)\}\!\!\}\Big).$$

Using $\xi_{(y \to x)} = \zeta(\delta(y, x)) = \zeta(\delta_{ij})$ and the induction hypothesis $\widetilde{h}_y^{(l)} = h_y^{(l)}$, we obtain, *elementwise* inside each $N_r^+(x)$,

$$\widetilde{\Phi}^{(l)}(\widetilde{h}_y^{(l)}, \xi_{(y \to x)}) = \Phi^{(l)}(\Xi(h_y^{(l)}, \zeta(\delta_{ij}))).$$

Now distinguish the two TGNN cases:

- *Global (Eq. (4)).* Here $y = (u, t_i)$ is precisely the embedding source used by the TGNN, so

$$m_{r,+}^{(l)}(x) = \Omega_a\Big( \{\!\{ \Phi^{(l)}(\Xi(h_{(u,t_i)}^{(l)}, \zeta(\delta_{ij}))) \mid (u, t_i) \in N_r^+(x) \}\!\} \Big).$$

- *Local (Eq. (4)).* By construction of $K_{\mathrm{loc}}(TG)$, $N_r^+(x)$ contains the current-time counterparts $(u, t_j)$; hence

$$m_{r,+}^{(l)}(x) = \Omega_a\Big( \{\!\{ \Phi^{(l)}(\Xi(h_{(u,t_j)}^{(l)}, \zeta(\delta_{ij}))) \mid (u, t_i) \in N_r^+(x) \}\!\} \Big).$$

Summing over lags and using the disjoint union $N(v, t_j) = \biguplus_r N_r^+(x)$ yields

$$\underbrace{\Omega_a\Big( \{\!\{ \Phi^{(l)}(\Xi(\cdot, \zeta(\delta_{ij}))) \mid (u, t_i) \in N(v, t_j) \}\!\} \Big)}_{\text{TGNN } m_{t_j}^{(l)}(v)} = \underbrace{\Omega_a\Big( \biguplus_r \{\!\{ m_{r,+}^{(l)}(x) \}\!\} \Big),}_{\text{RGNN combined incoming}}$$

i.e., the TGNN aggregate $m_{t_j}^{(l)}(v)$ equals the RGNN's combined incoming per-lag message at $x$.

Finally, both models apply the *same* $(\Omega_c, \Gamma^{(l)})$ to $\{\!\{ h_{(v,t_j)}^{(l)} \}\!\}$ and the above aggregate, hence

$$h_{(v,t_j)}^{(l+1)} = \Gamma^{(l)}\Big( \Omega_c(\{\!\{ h_{(v,t_j)}^{(l)} \}\!\} \uplus \{\!\{ m_{t_j}^{(l)}(v) \}\!\}) \Big) = \Gamma^{(l)}\Big( \Omega_c(\{\!\{ h_{(v,t_j)}^{(l)} \}\!\} \uplus \biguplus_r \{\!\{ m_{r,+}^{(l)}(x) \}\!\}) \Big) = \widetilde{h}_{(v,t_j)}^{(l+1)}.$$

This proves the induction step and concludes the proof. $\qquad\qquad\qquad\qquad\qquad\qquad\qquad\qquad\qquad\quad\square$

# E. Optimization of Expressive Lottery Tickets

We now discuss the impact of the expressivity of sparse RGNN initializations with fixed pruning masks on a fixed and finite dataset on their optimization behavior assuming standard small-step weight updates. Throughout, we use the directed multi-relational RGNN layer from Eqs. (1)–(3) with: (i) single-layer linear maps (i.e., without bias) plus nonlinearity for all message and combine functions $\Phi_{r,\pm}^{(l)}$ and $\Gamma^{(l)}$, (ii) a real-analytic activation $\sigma$ that is continuously differentiable, injective, satisfies $\sigma(0) = 0$, and $\sigma'(x) \neq 0$ for all $x$, (iii) sum aggregation/combination as in Section 2, (iv) a fixed finite dataset $\mathcal{D}$ of finite input graphs with labels $y$. We focus on graph classification for clarity; node-level tasks are analogous.

Let $\Theta$ denote the collection of all trainable RGNN parameters, including all weights of the message maps $\Phi_{r,\pm}^{(l)}$ and the combine maps $\Gamma^{(l)}$. Let $\Theta_k$ denote the collection at optimization step $k$. We index individual parameters by $\Theta_{k,i}$. A *mask* $M$ is a binary tensor of the same shape as $\Theta$, and a sparse initialization is given by

$$\Theta_0 = M \odot \widetilde{\Theta}_0,$$

for some base initialization $\widetilde{\Theta}_0$, with surviving parameter indices $\mathrm{supp}(M) = \{i \mid M_i = 1\}$. Let $L \in \mathbb{N}$ denote the network depth. A mask $M$ is *expressive* for $\mathcal{D}$ up to depth $L$ if there exists a parameter setting $\widehat{\Theta}$ with $\mathrm{supp}(\widehat{\Theta}) \subseteq \mathrm{supp}(M)$ such that for all $G_a, G_b \in \mathcal{D}$,

$$G_a \not\simeq_{\mathrm{RWL}^{(L)}} G_b \implies \Psi_{\widehat{\Theta}}^{(L)}(G_a) \neq \Psi_{\widehat{\Theta}}^{(L)}(G_b),$$

i.e., the masked RGNN matches the $L$-round 1-RWL distinguishing power on $\mathcal{D}$. An expressive mask $M$ is *irreducible* if for every $i \in \mathrm{supp}(M)$, the mask $M^{(-i)}$ obtained by setting $M_i = 0$ is not expressive for $\mathcal{D}$ up to depth $L$. Irreducibility captures the intuition of a "minimal expressive ticket": every surviving parameter is needed (on $\mathcal{D}$) to maintain the desired expressivity.

Let $y(G)$ denote the label of graph $G \in \mathcal{D}$. Relational 1-WL induces an equivalence relation $\equiv_{\mathrm{RWL}^{(L)}}$ on graphs via $L$ refinement steps. Within our RGNN class, $\equiv_{\mathrm{RWL}^{(L)}}$ is the fundamental upper bound on what can be distinguished. We are interested only in those WL-distinctions that are *label-relevant* on $\mathcal{D}$. A pair $(G_a, G_b) \in \mathcal{D}^2$ with $G_a \neq G_b$ is *task-relevant* (for depth $L$) if

$$G_a \not\equiv_{\mathrm{RWL}^{(L)}} G_b \quad \text{and} \quad y(G_a) \neq y(G_b).$$

That is, the architecture *can* distinguish $G_a, G_b$ at depth $L$, and the labels *require* it.

Let $M$ be a binary mask over the parameters of an $L$-layer RGNN. We call $M$ *task-expressive* for $(\mathcal{D}, y)$ up to depth $L$ if there exists a parameter setting $\widehat{\Theta}$ with $\mathrm{supp}(\widehat{\Theta}) \subseteq \mathrm{supp}(M)$ such that for all task-relevant pairs $(G_a, G_b)$,

$$\Psi_{\widehat{\Theta}}^{(L)}(G_a) \neq \Psi_{\widehat{\Theta}}^{(L)}(G_b).$$

In other words: the masked network realizes all label-relevant distinctions that are representable by $L$-round 1-RWL. A task-expressive mask $M$ is *irreducible* if for every $i \in \mathrm{supp}(M)$, the mask $M^{(-i)}$ obtained by setting $M_i = 0$ is *not* task-expressive for $(\mathcal{D}, y)$ up to depth $L$. Irreducibility in this context means: every remaining parameter is needed to realize *some* required label-relevant distinction on $\mathcal{D}$ (given the architectural 1-RWL limit).

We next formalize when a surviving parameter structurally participates in the supervised computation. Let $\mathcal{L}(\Theta)$ be the empirical loss on $(\mathcal{D}, y)$ under parameters $\Theta$:

$$\mathcal{L}(\Theta) \;=\; \frac{1}{|D|} \sum_{G \in \mathcal{D}} \ell(\Psi_{\Theta}^{(L)}(G),\, y(G)),$$

with some per-graph loss function $\ell(\cdot, \cdot)$ (e.g., cross-entropy).

Let $M$ be a mask and consider an RGNN with parameters supported on $M$ and loss $\mathcal{L}$ on $(\mathcal{D}, y)$. A parameter index $i \in \mathrm{supp}(M)$ is *gradient-connected (w.r.t. $(\mathcal{D}, y)$)* if there exists some $G \in \mathcal{D}$ and some output coordinate $j$ such that:

(i) the forward computation from $G$ to that output depends on $\Theta_{k,i}$ via a path of operations through the weights of some $\Phi_{r,\pm}^{(l)}$ and/or $\Gamma^{(l)}$,

(ii) along this path, all local derivatives (including $\sigma'$) are nonzero on the realized inputs.

Equivalently, $\partial\ell(\Psi_{\Theta_k}^{(L)}(G), y(G))/\partial\Psi_{\Theta_{k,i}}^{(L)}(G)_j \neq 0$ for some $G \in \mathcal{D}$ and $j$. A non-connected parameter is structurally dead for this task on $\mathcal{D}$: changing it never affects any prediction on $\mathcal{D}$.

Using the RGNN equations and the chain rule, we now spell out the explicit gradients for the linear $+\sigma$ blocks $\Gamma^{(l)}$ and $\Phi_{r,\pm}^{(l)}$ in the directed multi-relational case (Eqs. (1)–(3)), with sum aggregation/combination[3].

We parameterize

$$\Phi_{r,\pm}^{(l)}(h) = \sigma(W_{r,\pm}^{(l)} h), \qquad \Gamma^{(l)}(x) = \sigma(W_{\Gamma}^{(l)} x), \tag{11}$$

where $\sigma$ is injective, continuously differentiable, satisfies $\sigma(0) = 0$ and $\sigma'(x) \neq 0$ for all $x$, and $W_{r,\pm}^{(l)}, W_{\Gamma}^{(l)} \in \mathbb{R}^{d \times d}$.

Instantiating $\Omega_a$ as the sum, for each $v \in V$ and $r \in \mathcal{R}$, the per-relation messages are

$$m_{r,-}^{(l)}(v) = \sum_{u \in N_r^-(v)} \Phi_{r,-}^{(l)}(h_u^{(l)}), \qquad m_{r,+}^{(l)}(v) = \sum_{u \in N_r^+(v)} \Phi_{r,+}^{(l)}(h_u^{(l)}). \tag{12}$$

Instantiating $\Omega_c$ as the sum, the input to $\Gamma^{(l)}$ is

$$a_v^{(l)} = h_v^{(l)} + \sum_{r \in \mathcal{R}} m_{r,-}^{(l)}(v) + \sum_{r \in \mathcal{R}} m_{r,+}^{(l)}(v), \tag{13}$$

and

$$z_v^{(l+1)} \;=\; W_{\Gamma}^{(l)} a_v^{(l)}, \qquad h_v^{(l+1)} \;=\; \sigma(z_v^{(l+1)}). \tag{14}$$

Let $\mathcal{L}$ be the training loss on the fixed finite dataset $\mathcal{D}$. All sums below range over all graphs in $\mathcal{D}$ and their nodes.

We define the local backpropagation factor at layer $l+1$:

$$\delta_v^{(l+1)} \;:=\; \frac{\partial \mathcal{L}}{\partial z_v^{(l+1)}} \;=\; \frac{\partial \mathcal{L}}{\partial h_v^{(l+1)}} \odot \sigma'(z_v^{(l+1)}). \tag{15}$$

---

[3]As sanity-check, we applied our derived formulas and numerically compared the obtained gradients with those obtained via PyTorch's autograd: https://github.com/SamirMoustafa/gcn-rgcn-gradient-verification

**Gradients for $\Gamma^{(l)}$.** By the chain rule

$$\frac{\partial \mathcal{L}}{\partial W_\Gamma^{(l)}} = \sum_v \delta_v^{(l+1)} (a_v^{(l)})^\top. \tag{16}$$

The gradient w.r.t. $a_v^{(l)}$ is

$$\frac{\partial \mathcal{L}}{\partial a_v^{(l)}} = (W_\Gamma^{(l)})^\top \delta_v^{(l+1)} =: g_v^{(l)}. \tag{17}$$

Since $a_v^{(l)}$ is a sum of $h_v^{(l)}$ and all $m_{r,\pm}^{(l)}(v)$ (see (13)), each summand receives the same contribution:

$$\frac{\partial \mathcal{L}}{\partial m_{r,+}^{(l)}(v)} = g_v^{(l)}, \qquad \frac{\partial \mathcal{L}}{\partial m_{r,-}^{(l)}(v)} = g_v^{(l)}, \tag{18}$$

and $g_v^{(l)}$ also contributes to $\partial \mathcal{L}/\partial h_v^{(l)}$ via the self-term.

**Gradients for $\Phi_{r,+}^{(l)}$.** For each $r$ and $u$, define

$$z_{r,+}^{(l)}(u) = W_{r,+}^{(l)} h_u^{(l)}, \qquad \phi_{r,+}^{(l)}(u) = \Phi_{r,+}^{(l)}(h_u^{(l)}) = \sigma(z_{r,+}^{(l)}(u)). \tag{19}$$

Since $m_{r,+}^{(l)}(v) = \sum_{u \in N_r^+(v)} \phi_{r,+}^{(l)}(u)$, we obtain

$$\frac{\partial \mathcal{L}}{\partial \phi_{r,+}^{(l)}(u)} = \sum_{v: u \in N_r^+(v)} \frac{\partial \mathcal{L}}{\partial m_{r,+}^{(l)}(v)} = \sum_{v: u \in N_r^+(v)} g_v^{(l)}. \tag{20}$$

We define

$$\delta_{r,+}^{(l)}(u) := \frac{\partial \mathcal{L}}{\partial z_{r,+}^{(l)}(u)} = \frac{\partial \mathcal{L}}{\partial \phi_{r,+}^{(l)}(u)} \odot \sigma'(z_{r,+}^{(l)}(u)), \tag{21}$$

which is nonzero whenever the upstream terms are nonzero, since $\sigma'(x) \neq 0$. Then

$$\frac{\partial \mathcal{L}}{\partial W_{r,+}^{(l)}} = \sum_u \delta_{r,+}^{(l)}(u) (h_u^{(l)})^\top. \tag{22}$$

**Gradients for $\Phi_{r,-}^{(l)}$.** The derivation is analogous for the incoming-direction block. For each $r$ and $u$,

$$z_{r,-}^{(l)}(u) = W_{r,-}^{(l)} h_u^{(l)}, \qquad \phi_{r,-}^{(l)}(u) = \sigma(z_{r,-}^{(l)}(u)), \tag{23}$$

and since $m_{r,-}^{(l)}(v) = \sum_{u \in N_r^-(v)} \phi_{r,-}^{(l)}(u)$,

$$\frac{\partial \mathcal{L}}{\partial \phi_{r,-}^{(l)}(u)} = \sum_{v: u \in N_r^-(v)} \frac{\partial \mathcal{L}}{\partial m_{r,-}^{(l)}(v)} = \sum_{v: u \in N_r^-(v)} g_v^{(l)}, \tag{24}$$

$$\delta_{r,-}^{(l)}(u) := \frac{\partial \mathcal{L}}{\partial z_{r,-}^{(l)}(u)} = \frac{\partial \mathcal{L}}{\partial \phi_{r,-}^{(l)}(u)} \odot \sigma'(z_{r,-}^{(l)}(u)), \tag{25}$$

and thus

$$\frac{\partial \mathcal{L}}{\partial W_{r,-}^{(l)}} = \sum_u \delta_{r,-}^{(l)}(u) (h_u^{(l)})^\top. \tag{26}$$

These explicit expressions specialize the backpropagation rules to the RGNN blocks $\Gamma^{(l)}$ and $\Phi_{r,\pm}^{(l)}$ with sum aggregation, and are exactly the gradients used in our optimizability analysis.

**Lemma E.1.** *Let $M$ be a mask and let $i \in \text{supp}(M)$ be gradient-connected for the RGNN described in Eqs. (11)–(26). Then, for almost all initializations $\Theta_0$ supported on $M$, and for any nonzero loss gradient at the network outputs on some $G \in \mathcal{D}$, we have*

$$\frac{\partial \mathcal{L}}{\partial \Theta_{0,i}}(\Theta_0) \neq 0.$$

*Proof.* By construction of the RGNN layer, every trainable parameter in $\Theta$ belongs to one of the matrices $W_\Gamma^{(l)}$ or $W_{r,\pm}^{(l)}$. We show that for a gradient-connected parameter, the corresponding partial derivative is a nontrivial smooth function of $\Theta$, hence nonzero almost everywhere.

**Case 1:** $\Theta_i$ is an entry of $W_\Gamma^{(l)}$.

From (16), each entry of $\partial \mathcal{L} / \partial W_\Gamma^{(l)}$ is of the form

$$\left[\frac{\partial \mathcal{L}}{\partial W_\Gamma^{(l)}}\right]_{pq} = \sum_v \delta_v^{(l+1)}[p] \, a_v^{(l)}[q],$$

where $\delta_v^{(l+1)}$ is defined in (15) and $a_v^{(l)}$ in (13). Gradient-connectedness of $\Theta_i$ means there exists at least one node $v$ and output coordinate such that: (i) $a_v^{(l)}[q]$ depends (through the RGNN computation) on earlier features along a path using $\Theta_i$, and (ii) the upstream factor $\delta_v^{(l+1)}[p]$ receives nonzero signal from the loss whenever the output loss gradient is nonzero. All involved mappings are compositions of linear transformations and $\sigma$ with $\sigma'(x) \neq 0$, hence are smooth and locally invertible along the path.

Consequently, the function $\Theta \mapsto [\partial \mathcal{L} / \partial W_\Gamma^{(l)}]_{pq}$ is not identically zero on the parameter space restricted by $M$: at some admissible parameter setting, the corresponding summand $\delta_v^{(l+1)}[p] a_v^{(l)}[q]$ is nonzero. A nontrivial real-analytic (indeed smooth) function has a zero set of Lebesgue measure zero. Thus for almost all $\Theta_0$ supported on $M$, $[\partial \mathcal{L} / \partial W_\Gamma^{(l)}]_{pq}(\Theta_0) \neq 0$.

**Case 2:** $\Theta_i$ is an entry of $W_{r,+}^{(l)}$.

From (22),

$$\left[\frac{\partial \mathcal{L}}{\partial W_{r,+}^{(l)}}\right]_{pq} = \sum_u \delta_{r,+}^{(l)}(u)[p] \, h_u^{(l)}[q],$$

with $\delta_{r,+}^{(l)}(u)$ given by (21). Each $\delta_{r,+}^{(l)}(u)$ is obtained from upstream gradients $g_v^{(l)}$ (see (17), (20)) multiplied elementwise by $\sigma'(z_{r,+}^{(l)}(u))$, which is never zero. If $\Theta_{0,i}$ is gradient-connected, there exists at least one path from $\Theta_{0,i}$ to some output coordinate through $h_u^{(l)} \mapsto z_{r,+}^{(l)}(u) \mapsto \phi_{r,+}^{(l)}(u) \mapsto m_{r,+}^{(l)}(v) \mapsto a_v^{(l)} \mapsto h^{(l+1)} \mapsto \ldots$ that is used in the computation on some $G \in \mathcal{D}$. Along this path, all local derivatives are nonzero by assumption, and the loss provides a nonzero upstream signal. Therefore the corresponding summand $\delta_{r,+}^{(l)}(u)[p] h_u^{(l)}[q]$ is a non-constant smooth function of $\Theta$ and is not identically zero. As in Case 1, its zero set has measure zero, hence for almost all $\Theta_0$,

$$\left[\frac{\partial \mathcal{L}}{\partial W_{r,+}^{(l)}}\right]_{pq}(\Theta_0) \neq 0.$$

**Case 3:** $\Theta_i$ is an entry of $W_{r,-}^{(l)}$.

This is identical to Case 2, using Eqs. (23)–(26): for a gradient-connected entry, at least one term $\delta_{r,-}^{(l)}(u)[p] h_u^{(l)}[q]$ is a nontrivial smooth function of $\Theta$, hence nonzero for almost all $\Theta_0$.

In all cases, if $i$ is gradient-connected, the map $\Theta \mapsto \partial \mathcal{L} / \partial \Theta_i$ is not identically zero on the parameter space compatible with $M$. Thus its zero set is a measure-zero subset, and for almost all initializations $\Theta_0$ supported on $M$ we obtain $\partial \mathcal{L} / \partial \Theta_{0,i}(\Theta_0) \neq 0$ whenever the loss gradient at the outputs is nonzero. $\square$

A sparse initialization $\Theta_0$ with mask $M$ is *optimizable on* $(\mathcal{D}, y)$ if:

(i) every $i \in \mathrm{supp}(M)$ is gradient-connected w.r.t. $(\mathcal{D}, y)$,

(ii) for almost all $\Theta_0$ supported on $M$, all surviving parameters receive nonzero gradient at initialization, i.e., $\frac{\partial \mathcal{L}}{\partial \Theta_{0,i}}(\Theta_0) \neq 0$ whenever the loss gradient at the outputs is nonzero,

(iii) thus small gradient steps from $\Theta_0$ can affect all task-relevant directions permitted by $M$ (no structurally dead sub-blocks).

**Proposition E.2.** *Let $M$ be an irreducible task-expressive mask for $(\mathcal{D}, y)$ up to depth $L$ in the RGNN architecture of Eqs. (11)–(26). Then every $i \in \mathrm{supp}(M)$ is gradient-connected w.r.t. $(\mathcal{D}, y)$. Consequently, for almost all initializations $\Theta_0$ supported on $M$, $\Theta_0$ is optimizable on $(\mathcal{D}, y)$.*

*Proof.* Suppose, for contradiction, that there exists $i \in \mathrm{supp}(M)$ that is not gradient-connected. By definition, "not gradient-connected" is equivalent to

$$\frac{\partial \Psi_{\Theta}^{(L)}(G)}{\partial \Theta_{0,i}} \equiv 0 \quad \text{for all } G \in \mathcal{D},$$

i.e., the RGNN outputs on $\mathcal{D}$ are independent of $\Theta_{0,i}$. Because the loss $\mathcal{L}$ is computed from these outputs, the chain rule together with the explicit gradients (16), (22), (26) implies

$$\frac{\partial \mathcal{L}}{\partial \Theta_{0,i}}(\Theta) \equiv 0 \quad \text{for all parameter settings } \Theta \text{ supported on } M.$$

Equivalently, varying $\Theta_{0,i}$ while keeping all other coordinates fixed never changes any prediction on $\mathcal{D}$.

Now consider the reduced mask $M^{(-i)}$ obtained from $M$ by setting its $i$-th entry to zero. For any $\Theta$ supported on $M$, define $\Theta'$ supported on $M^{(-i)}$ by

$$\Theta'_{0,j} = \begin{cases} \Theta_{0,j}, & j \neq i, \\ 0, & j = i. \end{cases}$$

Since the outputs on $\mathcal{D}$ are independent of $\Theta_i$, we have

$$\Psi_{\Theta}^{(L)}(G) = \Psi_{\Theta'}^{(L)}(G) \quad \text{for all } G \in \mathcal{D}.$$

Thus $M^{(-i)}$ can realize exactly the same functions on $(\mathcal{D}, y)$ as $M$.

Because $M$ is task-expressive, there exists some $\widehat{\Theta}$ supported on $M$ that realizes all required label-relevant distinctions By the above argument, zeroing out $\widehat{\Theta}_{0,i}$ yields a parameter $\widehat{\Theta}'$ supported on $M^{(-i)}$ that induces the same predictions on $\mathcal{D}$, and hence $M^{(-i)}$ is also task-expressive. This contradicts irreducibility , which requires that removing any $i \in \mathrm{supp}(M)$ destroys task-expressivity.

Therefore, every $i \in \mathrm{supp}(M)$ must be gradient-connected.

Finally, by Lemma E.1, for each gradient-connected $i \in \mathrm{supp}(M)$ the partial derivative $\partial \mathcal{L} / \partial \Theta_{0,i}$ is nonzero for almost all initializations $\Theta_0$ supported on $M$ whenever there is a nonzero loss gradient at the outputs on some $G \in \mathcal{D}$. Since this holds for all surviving parameters, $\Theta_0$ is optimizable on $(\mathcal{D}, y)$. $\square$

**Proposition E.3.** *Fix depth $L$ and a sparsity level $s$. There exist masks $M'$ with $|\mathrm{supp}(M')| = s$ such that:*

(a) *$M'$ is not task-expressive for $(\mathcal{D}, y)$ up to depth $L$, i.e., it fails to separate some task-relevant pair $(G_a, G_b)$,*

(b) *some $i \in \mathrm{supp}(M')$ is not gradient-connected w.r.t. $(\mathcal{D}, y)$, so $\frac{\partial \mathcal{L}}{\partial \Theta_{k,i}}(\Theta) = 0$ for all $\Theta$ and all training steps.*

*Thus $M'$ is strictly dominated by an irreducible task-expressive mask: it is worse both in terms of expressivity for the task and in terms of optimizability.*

*Proof.* Start from an irreducible task-expressive mask $\overline{M}$ and:

(i) remove some nonzeros on paths (through parameters of $\Phi_{r,\pm}^{(l)}$ or $\Gamma^{(l)}$) needed to separate a task-relevant pair $(G_a, G_b)$, and

(ii) reassign the same number of nonzeros to parameters that only affect intermediate features which are structurally disconnected from the loss. Concretely, pick a feature channel (or neuron) whose entire set of outgoing weights to subsequent layers and to the readout is pruned by $M'$, so that no computational path from this channel can reach a labeled output on $\mathcal{D}$. Place the new nonzeros on the incoming weights into this channel. By construction, these parameters lie in $\text{supp}(M')$ but any change to them can never influence the network outputs on $\mathcal{D}$, hence $\frac{\partial \mathcal{L}}{\partial \Theta_{k,i}}(\Theta) = 0$ for all $\Theta$.

The resulting mask $M'$ has the same sparsity but cannot realize all label-relevant distinctions anymore, so it is not task-expressive. By construction, the newly added parameters never lie on a path to the loss and hence are not gradient-connected, with identically zero gradients. $\square$

### E.1. Persistence Under Training

We now extend Proposition E.2 from initialization ($k=0$) to the whole optimization trajectory under small-step gradient-based updates with a fixed mask $M$. We write write a generic first-order update as

$$\Theta_{k+1} \ = \ \Theta_k + U_k(\Theta_k, \nabla \mathcal{L}(\Theta_k)), \tag{27}$$

where $U_k$ is continuous in its arguments and satisfies $\|U_k(\Theta_k, \nabla\mathcal{L}(\Theta_k))\| \le \kappa_k \|\nabla\mathcal{L}(\Theta_k)\|$ for some stepsize $\kappa_k > 0$ (e.g., gradient descent, SGD, momentum, Adam with bounded step). The mask $M$ remains fixed throughout.

We begin by showing that, under our construction, parameter gradients that are non-zero at a point $\Theta_k$ remain non-zero in a local neighbourhood of $\Theta_k$ in parameter space:

**Lemma E.4.** *Fix $(\mathcal{D}, y)$ and the RGNN block structure in Eqs. (11)–(26). Suppose that at iteration $k$ the output loss gradient is nonzero and for all $i \in \text{supp}(M)$ we have $\frac{\partial \mathcal{L}}{\partial \Theta_{k,i}}(\Theta_k) \ne 0$. Then there exists a radius $\omega_k > 0$ such that for every $\Theta$ with $\|\Theta - \Theta_k\| < \omega_k$,*

$$\frac{\partial \mathcal{L}}{\partial \Theta_i}(\Theta) \ne 0 \quad \text{for all } i \in \text{supp}(M).$$

*Proof.* For each surviving parameter index $i \in \text{supp}(M)$ define the scalar function

$$g_i(\Theta) \ := \ \frac{\partial \mathcal{L}}{\partial \Theta_i}(\Theta).$$

By construction of the RGNN layer, all forward computations (14), (19), (23) are obtained from the parameters $\Theta$ by finitely many compositions of

- linear maps (matrix–vector products, sums), and
- the activation $\sigma$, which is real-analytic and hence $C^1$,

together with finite sums over nodes and relations. The explicit gradient expressions (16), (17), (22), (26) are built from these quantities using again only sums, products, and $\sigma'$. All of these operations are continuous, and finite compositions and sums of continuous functions are continuous. Therefore, for each $i$, the map $g_i : \Theta \mapsto \partial\mathcal{L}/\partial\Theta_i(\Theta)$ is continuous on the parameter space (restricted by the mask $M$).

Next, fix some $i \in \text{supp}(M)$. By assumption, $g_i(\Theta_k) = \frac{\partial \mathcal{L}}{\partial \Theta_i}(\Theta_k) \ne 0$. Define

$$\varepsilon_i \ := \ \frac{1}{2}|g_i(\Theta_k)| \ > \ 0.$$

Using the standard definition of the continuity of a function, by continuity of $g_i$ at $\Theta_k$, there exists a radius $\omega_{k,i} > 0$ such that

$$\|\Theta - \Theta_k\| < \omega_{k,i} \quad \Longrightarrow \quad |g_i(\Theta) - g_i(\Theta_k)| < \varepsilon_i.$$

For any such $\Theta$ we have

$$|g_i(\Theta)| \geq |g_i(\Theta_k)| - |g_i(\Theta) - g_i(\Theta_k)| > |g_i(\Theta_k)| - \varepsilon_i = \varepsilon_i > 0.$$

Hence $g_i(\Theta) \neq 0$ for all $\Theta$ in the ball $B(\Theta_k, \omega_{k,i})$.

The mask support $\text{supp}(M)$ is finite, so we have only finitely many radii $\{\omega_{k,i}\}_{i \in \text{supp}(M)}$. Define

$$\omega_k := \min_{i \in \text{supp}(M)} \omega_{k,i} > 0.$$

If $\|\Theta - \Theta_k\| < \omega_k$, then $\|\Theta - \Theta_k\| < \omega_{k,i}$ for every $i \in \text{supp}(M)$, and by the previous step each $g_i(\Theta)$ is nonzero. Equivalently,

$$\frac{\partial \mathcal{L}}{\partial \Theta_i}(\Theta) = g_i(\Theta) \neq 0 \quad \text{for all } i \in \text{supp}(M),$$

whenever $\|\Theta - \Theta_k\| < \omega_k$. This is exactly the desired statement. $\qquad\square$

**Proposition E.5.** *Let $M$ be an irreducible task-expressive mask for $(\mathcal{D}, y)$ up to depth $L$ for the RGNN of Eqs. (11)–(26). Assume training follows (27) with a fixed mask $M$. Then, for almost all initializations $\Theta_0$ supported on $M$, the following holds:*

*(a) (Base case) At $k=0$, $\Theta_0$ is optimizable on $(\mathcal{D}, y)$ (Proposition E.2).*

*(b) (Inductive step) Suppose at some $k \geq 0$ the output loss gradient is nonzero on the current batch and $\Theta_k$ is optimizable. Let $\omega_k > 0$ be as in Lemma E.4. If the update satisfies $\|\Theta_{k+1} - \Theta_k\| < \omega_k$, then $\Theta_{k+1}$ is also optimizable.*

*Consequently, for any finite horizon $K \in \mathbb{N}$ there exists a stepsize schedule $\{\eta_0, \ldots, \eta_{K-1}\}$ such that $\Theta_k$ is optimizable for all $k \leq K$, provided the output loss gradient is nonzero at those iterations.*

*Proof.* (a) is exactly Proposition E.2.

For (b), assume $\Theta_k$ is optimizable. Then, by the definition of optimizability and the fixed-mask architecture, every $i \in \text{supp}(M)$ remains gradient-connected at all iterations: gradient-connectedness depends only on the existence of structural paths through $\Phi_{r,\pm}^{(l)}$ and $\Gamma^{(l)}$ together with $\sigma'(x) \neq 0$, not on the specific numerical values of $\Theta_k$. By the definition of optimizability and Lemma E.1, $\frac{\partial \mathcal{L}}{\partial \Theta_{k,i}}(\Theta_k) \neq 0$ for all surviving $i$, given a nonzero output loss gradient. Lemma E.4 yields a radius $\omega_k > 0$ within which these partial derivatives remain nonzero for all $i \in \text{supp}(M)$.

If the update obeys $\|\Theta_{k+1} - \Theta_k\| < \omega_k$, then $\frac{\partial \mathcal{L}}{\partial \Theta_{k+1,i}}(\Theta_{k+1}) \neq 0$ for all surviving $i$ whenever the output loss gradient is nonzero at iteration $k+1$. Thus items (i) and (ii) of the definition of optimizability hold at $k+1$, so $\Theta_{k+1}$ is optimizable. By induction, repeating this argument up to any finite $K$ proves the consequence. $\qquad\square$

# F. Computational cost model

For an MLP of depth $M$ (number of hidden layers), width $m$ (hidden dimension), and unstructured sparsity level $\rho \in [0, 1]$ applied to $n_{\text{in}}$ distinct inputs (e.g., nodes or branch inputs), we roughly approximate the per-forward MADDS as $\text{MADD}_{\text{MLP}}(n_{\text{in}}, m, M, \rho) = n_{\text{in}} \cdot M \cdot (1 - \rho) m^2 \approx N_{\max} \cdot M \cdot (1 - \rho) m_{\min}^2$. Here $m^2$ counts the weights in a dense $m \times m$ layer, $(1 - \rho)m^2$ is the number of surviving weights after pruning, and the factor $n_{\text{in}}$ accounts for applying the MLP to all inputs.

# G. Toy dataset and RGNN architecture

This appendix details the exact RGNN architecture and the synthetic multi-relational dataset generator used for Figure 2.

**Synthetic multi-relational dataset.** Each graph is a directed multi-relational graph $G = (V, \{E_r\}_{r \in \mathcal{R}})$, with $|V| = N$ nodes and $|\mathcal{R}| = R$ relations. We construct $E_r$ by first sampling an *undirected* Erdős–Rényi edge set $\tilde{E}_r \subseteq \{\{u, v\} \mid u < v, \ u, v \in V\}$ where each unordered pair is included independently with probability $p_r$. To avoid degenerate empty relations, if $\tilde{E}_r = \emptyset$ we add one random undirected edge $\{u, v\}$ with $u \neq v$. We then convert each undirected edge to two directed edges, i.e., $E_r := \{(u, v), (v, u) \mid \{u, v\} \in \tilde{E}_r\}$. To encourage structural diversity among generated

graphs, we reject duplicates under a cheap signature heuristic: for each relation $r$, we compute the (out-)degree sequence $(\deg_r(v))_{v \in V}$, $\quad \deg_r(v) := |\{u \in V \mid (v, u) \in E_r\}|$, sort it, and concatenate it with the undirected edge count $|\tilde{E}_r|$. A candidate graph is accepted only if its signature has not been seen before. Unless stated otherwise, we generate $G_{\text{set}} = 30$ graphs with $N = 6$ nodes and $R = 3$ relations using probabilities $(p_r)_{r=1}^{R} = (0.12, 0.20, 0.28)$ and a fixed random seed.

**Node features.** We use simple structural node features derived from degrees:

$$x_v := [1, \deg_{r_1}(v), \deg_{r_2}(v), \ldots, \deg_{r_R}(v)]^\top \in \mathbb{R}^{1+R}. \tag{28}$$

Thus the input dimension is $d_{\text{in}} = 1 + R$.

**RGNN architecture.** We instantiate a minimal directed multi-relational RGNN with $L$ layers, hidden width $\mathcal{D}$, and activation $\sigma = \tanh$. All linear maps are bias-free. First, we project node features to hidden states:

$$h_v^{(0)} = \sigma(W_{\text{in}} x_v), \qquad W_{\text{in}} \in \mathbb{R}^{d \times d_{\text{in}}}. \tag{29}$$

For each layer $\ell \in \{0, \ldots, L-1\}$ and relation $r \in \mathcal{R}$, we compute a (directed) *sum* aggregation

$$a_{v,r}^{(\ell)} = \sum_{(u,v) \in E_r} h_u^{(\ell)} \in \mathbb{R}^d, \tag{30}$$

apply a relation-specific linear map, and pass through $\tanh$:

$$m_{v,r}^{(\ell)} = \sigma\left(W_r^{(\ell)} a_{v,r}^{(\ell)}\right), \qquad W_r^{(\ell)} \in \mathbb{R}^{d \times d}. \tag{31}$$

We then combine the current state with all relation messages via summation (rather than concatenation) and apply a shared combine transform:

$$s_v^{(\ell)} = h_v^{(\ell)} + \sum_{r \in \mathcal{R}} m_{v,r}^{(\ell)}, \qquad h_v^{(\ell+1)} = \sigma\left(W_{\text{c}}^{(\ell)} s_v^{(\ell)}\right), \tag{32}$$

with $W_{\text{c}}^{(\ell)} \in \mathbb{R}^{d \times d}$. Finally, we produce a graph-level embedding by sum readout:

$$e(G) = \sum_{v \in V} h_v^{(L)} \in \mathbb{R}^d. \tag{33}$$

In our experiments we use $d = 16$ or $8$ and $L = 3$ unless stated otherwise, and we evaluate embeddings without training (random initialization as in standard `nn.Linear` defaults).

# H. Assumptions and Limitations

**Assumptions.** Our results are stated and proved under the following assumptions, which are either inherent to the setting (finite datasets) or technically convenient for establishing dataset-conditional injectivity guarantees under random pruning.

All expressivity and probability guarantees are dataset-conditional: we fix a *finite* dataset $\mathcal{D}$ of *finite* (relational or temporal) graphs. This makes the sets of MLP inputs actually *witnessed* on $\mathcal{D}$ finite and allows us to bound non-injectivity events via combinatorial arguments.

The expressivity target is $L$-round 1-RWL (and its temporal lift via the corresponding knowledge-graph construction). Concretely, the theorems guarantee that, with probability at least $\gamma > 0$ over a random mask, a pruned network preserves the ability to separate all pairs in $\mathcal{D}$ that are separated by $L$ rounds of the corresponding 1-RWL refinement.

We analyze random pruning at initialization via a Bernoulli mask $M \sim \mathcal{B}_\rho$ (entries independently set to zero with probability $\rho$). We additionally assume a bounded base initialization (e.g., $\Theta_0 \sim \mathcal{U}_c^d$) and define the sparse initialization by $\widehat{\Theta}_0 = M \odot \Theta_0$.

Initial node labels are encoded by an injective map $\Lambda$, ensuring label collisions are not introduced at input.

The RWL-to-RGNN correspondence relies on injectivity of the multiset encoders and update maps *on the finite domain witnessed on $\mathcal{D}$*. The probability lower bounds are expressed in terms of (i) the maximum number $N_{\text{max}}$ of distinct witnessed

MLP inputs, (ii) the minimum $\ell_0$-separation $s_{\min}$ between distinct witnessed inputs, and (iii) the minimum hidden width $m_{\min}$ across the relevant MLP blocks.

For all MLPs (e.g., $\Gamma^{(l)}$ and, in the optimization discussion, also $\Phi_{r,\pm}^{(l)}$), we assume a real-analytic, injective, continuously differentiable, zero-fixing activation $\sigma$ with a nowhere-zero derivative (i.e., $\sigma(0) = 0$ and $\sigma'(x) \neq 0$ for all $x$).These assumptions are technical and simplify the analysis. They hold for smooth monotone activations such as $\tanh$ and *shifted sigmoids* (e.g., $x \mapsto \mathrm{sigmoid}(x) - \mathrm{sigmoid}(0)$, which enforces $\sigma(0) = 0$) but exclude non-smooth activations such as ReLU. Extending the theory to non-smooth activations is conceptually possible, but would require different lemmata and additional case distinctions, which we consider future work beyond the scope of the current paper.

In Appendix E we focus on the RGNN block structure induced by Eqs. (1)–(3) with single-layer linear maps (no bias) plus nonlinearity for all message and combine functions, sum aggregation/combination, a fixed finite dataset $(\mathcal{D}, y)$, and small-step first-order updates with a fixed mask.

**Limitations and scope.** The above assumptions enable clean, explicit probability bounds and a unifying reduction viewpoint, but they also delimit what we claim.

Theorems certify preservation of RWL-separations on a fixed finite dataset; they do not constitute a population-level or distributional generalization guarantee.

The guarantee is with respect to $L$-round 1-RWL (and the temporal analogue). It does not address distinctions beyond this baseline (e.g., higher-order WL tests) and does not, by itself, guarantee the best possible expected test-set performance for a given task.

The probability lower bounds are derived for independent Bernoulli pruning at initialization (and independence across MLP blocks when composing across layers). Structured pruning (e.g., magnitude-based, channel-wise) is outside the formal guarantee unless separately analyzed.

The lower bound $\gamma_{\mathrm{RGNN}}$ is derived via per-block non-injectivity bounds and union-bounding; as a result it can be conservative, particularly near the critical sparsity/width regime where expressivity transitions from holding to failing. Empirically, it tightens again in the high- and low-success regimes.

The certificate depends on $N_{\max}$ and $s_{\min}$ (witnessed input counts and separations). These quantities capture dataset/architecture interaction, but they can be large or difficult to control a priori; consequently, the bound may be pessimistic even when expressive subnetworks exist.

Appendix E establishes non-degeneracy/gradient-connectivity properties and local persistence of nonzero gradients under small-step updates for fixed masks; it is not a global convergence result.

