# OpenReview forum: "A Unifying Relational Perspective on Expressive Lottery Tickets"
_ICML.cc/2026/Conference — ICML 2026 spotlight_

### Official Review · Reviewer_6rrp · 2026-03-03

**Soundness:** 3
**Presentation:** 4
**Significance:** 3
**Originality:** 3
**Overall Recommendation:** 5
**Confidence:** 4

**Summary:**

The paper generalizes the strong expressive lottery ticket hypothesis (SELTH) to multi-relational and temporal graph message passing architectures. The authors prove that sufficiently overparameterized Relational GNNs (RGNNs) contain sparse subnetworks that preserve 1-RWL expressivity with positive probability under random pruning. They derive an explicit lower bound on this probability. Furthermore, they show that temporal GNNs and cross-graph MPNNs admit RGNN reformulations, allowing the expressivity analysis to extend to these architectures as well.

**Compliance With Llm Reviewing Policy:**

Affirmed.

**Final Justification:**

The rebuttal addressed most of my comments and thus I maintained by initial positive recommendation.

**Key Questions For Authors:**

1. The bound presented around line 227 is acknowledged to be relatively loose. Are there additional assumptions (e.g., on the masking distribution) that could make the bound tighter?

2. Could these results be extended to more expressive MPNN variants, for example, architectures corresponding to higher-order k-WL tests?

**Limitations:**

1. Experimental analysis is restrictive but sufficient given the theoretical contribution of this work.
2. The expressivity (or seperability) of sparse models is preserved on a given finite space of graphs and not uniformly across all graphs.

**Strengths And Weaknesses:**

## Strengths

- The paper extends SELTH to multi-relational and temporal graphs in a non-trivial way. Although the main argument builds on known results (notably Lemma A.1 from Kummer et al., 2025b), the combination and extension of these ideas are carefully executed and technically rigorous.

- Theorem 3.1 provides a clear unification of the proposed result with classical SELTH, which appears as a special case (as noted in line 257).

- Theorems, definitions, and notation are carefully introduced and consistently used throughout the paper. Making the paper readable despite its technical nature. The main theoretical results are accompanied by helpful intuitive explanations. For example, Theorem 3.1 is followed by a discussion of the lower bound on $\tilde{\gamma}_{\text{RGNN}}$, where each term of the bound is discussed separately.


## Weaknesses

- At the beginning of Section 3, the Bernoulli masking procedure is described somewhat unclear. The sentence “We assume that all binary pruning masks $M$ with sparsity $\rho$ are constructed such that each entry is an independent Bernoulli random variable” could be written more formally. A precise definition of the mask distribution would improve clarity, especially for readers less familiar with the literature.

- The paper uses the term expressivity, but most results concern distinguishing or separability power. It would be helpful to clarify whether these notions are intended to coincide in this setting or to cite work explicitly connecting expressivity to separability results (e.g., Geerts & Reutter, Expressiveness and Approximation Properties of Graph Neural Networks). A short discussion of how such results might extend to RGNNs or temporal GNNs could also be included as future work.

- The expressivity result for temporal GNNs is somewhat implicit. It is explained that it follows from Lemmas 3.3, 3.4, and Theorem 3.1, but no explicit main statement appears in the main text. Since there are three consecutive lemmas without a summarizing theorem, including an informal (or formal) statement would improve clarity.

- The motivation for sparsely parameterized GNNs is comparatively weaker than the technical depth of the contribution. While the paper shows that sparse GNNs can remain expressive and trainable, it does not clearly justify why sparsity is particularly needed in multi-relational and temporal settings. A stronger connection to concrete scalability or deployment challenges would make the motivation more compelling.

- Although the optimization analysis is carefully presented in the appendix, the main text does not clearly state what this section formally guarantees. Including a formal (or informal) statement would improve clarity and better highlight its contribution. This would also help address the broader motivation concerns mentioned above.

---

> ### Author Rebuttal · Authors · 2026-03-30
>
> We thank the reviewer for the insightful and positive review, especially for recognizing the non-trivial extension of SELTH, the unifying role of Theorem 3.1, and the paper’s effort to remain readable despite its technical depth. We address the comments and questions below.
>
> ## Comments
>
> **C1:** At the beginning of Section 3 [...]
>
> **A:** We agree that this should be stated more formally. As clarified in our reply to reviewer EkfA / Q2, our intended convention is that $\rho$ denotes the *sparsity ratio*, and we kindly refer you to this reply for details.
>
> **C2:** The paper uses the term expressivity [...]
>
> **A:** In this paper, we use *expressivity* in the WL/RWL sense, i.e., as *distinguishing power*: the ability to realize the distinctions induced by $1$-WL / $1$-RWL on the graph classes under consideration.
>
> In the empirical section, *separability* is the corresponding *finite-dataset proxy*: we ask whether the embeddings are distinct on the graphs observed in the dataset. Thus, separability is not a different concept here, but the dataset-level operationalization of expressivity. We will make this terminology more explicit in the revision and distinguish it more clearly from stronger geometric notions of separability.
>
> **C3:** The expressivity result for temporal GNNs is somewhat implicit [...]
>
> **A:** The current argument already establishes the result by combining Lemmas 3.3 and 3.4 with Theorem 3.1, but we agree that, after several consecutive lemmas, an explicit summarizing statement in the main text would improve readability.
>
> We will therefore add a short statement or corollary making explicit that overparameterized local/global TGNNs inherit the same probabilistic $1$-RWL expressivity-preservation guarantee via the TGNN$\to$RGNN reduction.
>
> **C4:** The motivation for sparsely parameterized GNNs is comparatively weaker [...]
>
> **A:** RGNNs can become parameter-heavy due to multiple relation-/direction-specific branches and transformations. For example, XIMP's parameter count scales quadratically with the number of abstractions. In such a setting, showing that substantial pruning can preserve expressivity and trainability already provides a principled justification for sparsification beyond the single-relation static case. Please see our reply to wg6V / Q1.
>
> We will strengthen this motivation in the revision, while clarifying that our present contribution is a foundational expressivity/optimization result.
>
> **C5:** Although the optimization analysis is carefully presented in the appendix, the main text [...]
>
> **A:** Informally, Appendix E shows that for irreducible task-expressive masks on a fixed finite dataset, all surviving parameters are gradient-connected and, for almost all supported initializations, receive nonzero gradients whenever the output loss gradient is nonzero; under sufficiently small first-order updates, this property is preserved during training.
>
> This guarantee is already summarized briefly in the main text, but we agree that a clearer standalone statement would better highlight its role. We will therefore make this summary more explicit and formal in the revision.
>
> ## Questions
>
> **Q1:** The bound presented around line 227 is acknowledged to be relatively loose [...]
>
> **A:** As noted in our reply to reviewer wg6V / C2, the present bound is conservative because it combines per-block injectivity bounds with union-bounding and then collapses to worst-case branch/combine statistics before composing across layers.
>
> Accordingly, tighter bounds should indeed be possible under stronger assumptions. Two natural directions are: (i) retaining block-specific quantities instead of replacing them by the worst-case values $\widetilde N_{\max}, \widetilde s_{\min}, \widetilde m_{\min}$, and (ii) analyzing more structured masking distributions than independent Bernoulli pruning at initialization. Our current choice of independent Bernoulli masking is deliberate, since it yields a clean and explicit closed-form guarantee. We will clarify this trade-off more explicitly in the revision.
>
> **Q2:** Could these results be extended to more expressive MPNN variants [...]
>
> **A:** In principle, yes. It would be interesting to extend the framework to more expressive settings such as $k$-WL, $k$-RWL, and higher-order temporal GNNs; in this sense, our current work can be viewed as a first-order step that may provide useful insight for such extensions.
>
> At the same time, this is not immediate. Our analysis is explicitly scoped to the $1$-RWL setting, and extending it would require both a suitable higher-order relational/temporal expressivity characterization and a substantially more involved probability analysis for the corresponding higher-order message and combination blocks. Thus, while this is a promising future direction, it is not something that can be added straightforwardly within the scope of the current paper. Please also see our reply to reviewer qnLq / Q3, where we address a closely related point.

---

> > ### Author Rebuttal · Reviewer_6rrp · 2026-04-02
> >
> > Thank the authors for their rebuttal. The authors have addressed all my questions clearly, and I appreciate the clarifications. I support acceptance, as the paper makes a solid and useful contribution. I will keep my score at 5/6 to reflect its relative positioning compared to works with stronger novelty and impact, rather than any unresolved issues.

---

### Official Review · Reviewer_EkfA · 2026-03-04

**Soundness:** 3
**Presentation:** 2
**Significance:** 3
**Originality:** 2
**Overall Recommendation:** 5
**Confidence:** 3

**Summary:**

The Strong Expressive Lottery Ticket Hypothesis (SELTH) is all about sparse subnetworks that preserve expressivity, intended as the capability of distinguishing non-isomorphic graphs. This work extends the SELTH to a broader class of graph architectures, including multi-relational (RGNN), temporal (TGNN), and cross-graph message-passing models. While previous literature has primarily focused on existence proofs for expressive tickets in static, single-relational GNNs, this paper derives a lower bound on the probability that a randomly pruned subnetwork retains 1-RWL expressivity on a finite dataset. This approach involves a reformulation of temporal and hierarchical architectures into a unified RGNN format. Furthermore, the work connects theory and practice by showing that expressivity of a sparse RGNN serves as a predictor of gradient flow and ultimate model performance.

**Compliance With Llm Reviewing Policy:**

Affirmed.

**Final Justification:**

The authors' rebuttal successfully addressed my concerns regarding their work. The novelty of providing a single framework for diverse pruning results is a significant contribution.

**Key Questions For Authors:**

1. How does the proposed "straightforward" extension to ReLU handle the loss of injectivity in a bias-free setting? Given that negative values are mapped to zero, it appears that distinct inputs would collapse into identical outputs at initialization. Does your theory implicitly assume the addition of bias terms, or is there another mechanism that prevents this expressivity collapse?
2. Clarification on the Bernoulli parameter $\rho$: In Section 3, you denote the mask distribution as $M \sim \mathbb{B}_\rho$, yet the surrounding text defines $\rho$ as the sparsity (the probability of an entry being zero, not one). Could you confirm if the entries in the mask is actually drawn from $M \sim \text{Bernoulli}(1 - \rho)$?

**Limitations:**

Yes.

**Strengths And Weaknesses:**

**Strengths**
1. The paper improves the current understanding of the SELTH by moving from existential proofs to more meaningful probabilistic guarantees. Rather than asking if an expressive subnetwork exists, the authors quantify how likely it is to find one through random pruning. They derive an explicit lower bound that accounts for network width and sparsity, effectively highlighting the conditions under which a pruned architecture remains 1-RWL expressive.
2. The authors do a great job of creating a unified framework for several GNN architectures, reframing models like temporal and hierarchical GNNs into a single relational framework. This allows them to provide a single generalized theoretical guarantee that covers a wide range of GNN variants under one umbrella.
3. By linking a subnetwork's ability to separate graphs at initialization to its final test performance, the authors provide a useful diagnostic for pruning. This connection shows that winning tickets are largely determined by their initial topological expressivity, highlighting that maintaining 1-RWL separation is a prerequisite for successful optimization in sparse regimes.
4. Code is provided for reproducibility of experiments.

**Weaknesses**

1. The authors claim that extending these results to other activations such as ReLU is straightforward. In the bias-free architecture analyzed in this study, ReLU’s zero-squashing property for negative pre-activations introduces a non-injectivity that would lead to feature collisions. Since the entire proof for 1-RWL preservation relies on injectivity, this does not look like a mere bookkeping issue, as noted in the paper.

2. The exposition of the theoretical results in the main paper makes it sometimes difficult to distinguish the paper’s original technical contributions from existing work. While Theorem 3.1 is presented as a primary result, the main text does not clearly delineate which parts of the proof rely on established lemmas from prior literature and which represent novel developments by the authors.


### Comments and typos:
- While the proof for Theorem 3.1 in Appendix D explicitly constrains the sparsity ratio to $\rho \in (0,1)$, this requirement is omitted from the theorem's statement (and surrounding text) in the main text. This should be clearly stated upfront to clarify that fully pruned or fully dense, i.e., $\rho=1$ and $\rho=0$, are not considered in the theorem.
- Hadamard is spelled with one 'd' (line 193, right column).
- There's an additional closed parentesis in equation (3).
- "Appenix H" at line 201 (right column) is missing a 'd'.
- The word Appendix is often mispelled as "Appenidx" throughout the document.
- "but with additional the constraint that" at line 289 (right column) should be "but with the addional constraint that".

---

> ### Author Rebuttal · Authors · 2026-03-30
>
> We thank the reviewer for their insightful and positive review, and in particular for recognizing the probabilistic strengthening of SELTH, the unifying relational framework across several GNN architectures, and the practical relevance of linking initial separability to downstream performance. We address the comments and questions below.
>
> ## Comments
>
> **C1:** The authors claim that extending these results to other activations such as ReLU is straightforward [...]
>
> **A:** The reviewer is correct that our current formal assumptions do *not* cover bias-free ReLU directly. Our remark about an extension to ReLU in Appendix H was intended only as a brief comment on possible extensions beyond the present formal scope, not as a claim that the current theorem already covers bias-free ReLU.
>
> We agree that the wording “straightforward” is too strong for what is currently established, and we will revise it accordingly.
>
> **C2:** The exposition of the theoretical results in the main paper makes it sometimes difficult to distinguish the paper’s original technical contributions from existing work [...]
>
> **A:** As noted in our reply to wg6V / C4, the only imported ingredient is a single-block lemma establishing injectivity of one masked MLP on a finite witnessed input set. The main contribution of Theorem 3.1 is then the lift of this ingredient to the relational multi-branch RGNN setting: connecting the RGNN layer structure to $1$-RWL, introducing the witnessed branch/combine input sets, and composing the argument across branches and layers to obtain the relational probability bound.
>
> We will revise the exposition in the main text and appendix to distinguish these inherited and novel components more explicitly.
>
> **C3:** Comments and typos.
>
> We thank the reviewer for their comments and will adjust the final revision accordingly.
>
> ## Questions
>
> **Q1:** How does the proposed "straightforward" extension to ReLU handle the loss of injectivity in a bias-free setting? [...]
>
> **A:** Our current theory does *not* implicitly assume bias terms, and, as clarified in our reply to C1, the present formal result does *not* cover bias-free ReLU. In that setting, the reviewer is correct that negative pre-activations can collapse distinct inputs to the same output, so the current injectivity-based proof does not carry over unchanged.
>
> The key point is that our theorem only requires injectivity on the *finite set of MLP inputs actually witnessed on the dataset*, not global injectivity on all of $\mathbb{R}^d$. Thus, a ReLU-based extension would not need to prevent all possible collisions everywhere; rather, it would require a separate finite-set injectivity argument showing that, under suitable additional conditions, these witnessed collisions are avoided with positive probability. So there is no hidden mechanism in the current proof that automatically rescues bias-free ReLU; instead, a ReLU extension would require a different lemma and additional case distinctions, which we consider future work beyond the scope of the current paper.
>
> We will clarify this explicitly in the revision.
>
> **Q2:** Clarification on the Bernoulli parameter $\rho$: [...]
>
> **A:** We thank the reviewer for bringing this to our attention. Our intended convention is that $\rho$ denotes the *sparsity ratio*, i.e.,
> $$
> \Pr[m_{ij}=0]=\rho, \qquad \Pr[m_{ij}=1]=1-\rho.
> $$
> So under the standard Bernoulli parameterization, each mask entry should indeed be written as
> $$
> m_{ij}\sim\mathrm{Bernoulli}(1-\rho).
> $$
> We agree that the notation $M\sim\mathscr{B}_\rho$ is ambiguous and will revise it to make the convention explicit.

---

> > ### Author Rebuttal · Reviewer_EkfA · 2026-04-03
> >
> > I am satisfied with the authors' rebuttal, and I will increase my score provided that no substantial problem emerges from the discussion with the other reviewers.

---

> > > ### Author Response · Authors · 2026-04-04
> > >
> > > We are very happy that our rebuttal fully addressed your comments!
> > >
> > > Considering that all other reviewers by now have acknowledged that our rebuttal fully addressed their comments as well and qnLq and wg6V have raised their scores from 4 to 5, we would like to kindly remind you  of your acknowledgement statement:
> > >
> > > "I will increase my score provided that no substantial problem emerges from the discussion with the other reviewers."
> > >
> > > and ask you to increase your score from 4 to 5 as you indicated.
> > >
> > > Thank you.

---

### Official Review · Reviewer_qnLq · 2026-03-11

**Soundness:** 3
**Presentation:** 3
**Significance:** 3
**Originality:** 3
**Overall Recommendation:** 5
**Confidence:** 2

**Summary:**

This paper focuses on expressive lottery ticket theory for relational and temporal graph neural networks.

A previous work, SELTH, proved the existence of expressive lottery tickets for static GNNs. This paper extends that theory to relational GNNs, replacing the 1-WL test with the 1-RWL test, and derives a probabilistic lower bound, $\gamma_{RGNN}$, for Relational SELTH. This bound depends on the maximum number of distinct inputs $N_{max}$, the minimum $l0$-separation $s_{min}$, the minimum hidden width of the MLPs $m_min$, the pruning probability $\rho$, the MLP depth $M$, the message-passing depth $L$, and the number of branches $\mathcal{B}$.

The paper further shows that both local and global temporal GNNs can be reformulated as relational GNNs, and therefore also satisfy the RSELTH theory. It additionally considers cross-graph message passing, showing that these architectures can also be transformed into relational GNNs. In this way, the paper provides a unified view of temporal GNNs, cross-graph message passing, and relational GNNs under the same theoretical framework.

Finally, the paper argues that after pruning, the resulting subnetwork is not only expressive, but also remains trainable under small-step first-order optimization methods.

**Compliance With Llm Reviewing Policy:**

Affirmed.

**Final Justification:**

The rebuttal fully resolves my concerns.

**Key Questions For Authors:**

1. See the weakness: regarding time stamps. More discussion would be nice.
2. I wonder how $L$ will affect the model. Since most GNNs can't have a large $L$, after a certain $L$, the $s_{min}$ will be very small (over-smoothing / overfitting issue). However, sometimes, GNNs need long-distance information passing, i.e., a large $L$. To this end, the probabilistic lower bound would be very loose. Any way to avoid this?
3. 1-RWL test compared with 1-WL test and k-WL test. A lot of GNNs are claimed to have the same expressive power as 2-WL test. More detailed comparison between 1-WL test, 2-WL test, and 1-RWL test can be included.

**Limitations:**

yes

**Strengths And Weaknesses:**

Strengths:

This paper extends SELTH to RSELTH and further provides a probabilistic lower bound.
The paper reformulates both TGNNs and cross-graph message passing as relational graphs, allowing them to share the theoretical guarantees of RSELTH.
The paper also argues that the pruned subnetwork remains trainable.
The presentation is clear and easy to follow.
The experimental results provide additional support for the theoretical claims.

Weaknesses:

However, the reformulation of TGNNs as relational graphs is somewhat delicate. From my understanding, this step relies on a rather strong assumption: the timestamps $t$, or the timestamp gaps $t_i - t_j$​, after encoding, must effectively map to a finite set. In many temporal graph settings, especially during testing and evaluation, this assumption may not hold. Timestamps can vary continuously, and even slight changes may produce new timestamp gaps. In that case, the number of induced relations could become extremely large, or even effectively unbounded, which may in turn make the probabilistic lower bound very weak in practice. I would appreciate it if the authors could clarify this point in more detail.

Another concern is the discussion of cross-graph message passing. The paper establishes a formal 1-RWL characterization for certain standardized cross-graph message-passing architectures, but it does not convincingly show that this 1-RWL characterization is sufficiently strong or practically explanatory for the empirical advantages of cross-graph message passing. Additional experiments or analysis would help strengthen this part.

---

> ### Author Rebuttal · Authors · 2026-03-30
>
> We thank the reviewer for the insightful and positive review, and in particular for recognizing the extension from SELTH to RSELTH with probabilistic guarantees, the unifying relational reformulation of TGNNs and cross-graph message passing, the trainability result for pruned subnetworks, and the effort to keep the presentation clear and experimentally well supported.
>
> ## Comments
>
> **C1:** However, the reformulation of TGNNs as relational graphs is somewhat delicate [...]
>
> **A:** Please see our reply to Q1, where we clarify this point in detail.
>
> **C2:** Another concern is the discussion of cross-graph message passing [...]
>
> **A:** Our contribution here is intentionally narrower. The purpose of the $1$-RWL characterization is to bring certain standardized cross-graph / hierarchical message-passing architectures into the same relational framework as RGNNs/TGNNs, so that the RSELTH guarantee transfers to them. It is not meant as a claim that $1$-RWL by itself fully explains the empirical advantages of cross-graph message passing.
>
> That said, the paper does provide initial empirical support that this perspective is practically meaningful: Figure 3 includes XIMP on ADMET and potency and shows that pre-training separability predicts test performance. We agree that more targeted cross-graph-specific analysis could strengthen this part further, and we will revise the text to state this scope more explicitly.
>
> ## Questions
> **Q1:** See the weakness: regarding time stamps. More discussion would be nice.
>
> **A:** Our main TGNN$\to$RGNN reduction does *not* turn each realized timestamp gap into a separate relation. In Lemma 3.3, the relation/direction branches remain those of $K_\star(TG)$, namely the finite lag relations $r=j-i$, while the real-valued gaps are stored as edge features $\xi_e:=\zeta(\delta(y,x))$ and processed inside the shared message map. Hence, continuously varying timestamps do not by themselves make the number of relations unbounded in the main construction.
>
> The relevant finiteness assumption is instead the same one already used in Theorem 3.1: the guarantee is for a finite dataset $D$ of finite graphs. Accordingly, for TGNNs we only require that finitely many time gaps are actually *witnessed on $D$*, which holds automatically when $D$ and each temporal graph in it are finite. We agree that this makes the result dataset-conditional rather than a distributional guarantee over arbitrary unseen continuous timestamps. The alternative construction that refines the relation alphabet by realized gaps can indeed become large, which is precisely why we place it in Appendix A and discuss its shortcomings there. We will make this distinction more explicit in the revision.
>
> **Q2:** I wonder how $L$ will affect the model [...]
>
> **A:** In our bound, $L$ enters through the exponent $L(M|\mathcal{B}|+1)$, so increasing depth makes the guarantee more demanding: at fixed width, $\widetilde \gamma_{\mathrm{RGNN}}$ decreases, and maintaining the same target probability requires either larger width or better separation. If increased propagation depth also reduces $s_{\min}$, then this further weakens the bound. Please also see our reply to reviewer wg6V / Q2, where we address a closely related point.
>
> At the same time, our theorem should be read as a conservative initialization-time guarantee on a finite dataset, not as a direct model of over-smoothing or overfitting during training. The connection to over-smoothing is therefore indirect, via its effect on separability. Within our framework, the main ways to counteract this are to increase width and to preserve larger input separation, e.g., through better encodings. More broadly, one may also reduce the need for very large $L$ by using architectures with more direct long-range communication pathways, such as cross-graph message passing. We will make this point clearer in the revision.
>
> **Q3:** 1-RWL test compared with 1-WL test and k-WL test [...]
>
> **A:** In our setting, $1$-RWL is the relational analogue of $1$-WL: it is the natural expressivity baseline for standard relation-aware message passing on multi-relational graphs, just as $1$-WL is for classic message-passing GNNs on static graphs. Indeed, when $|R|=1$ and directions are removed, our result reduces to the static $1$-WL / SELTH case.
>
> By contrast, $2$-WL is a higher-order test and is typically matched only by more expressive higher-order GNN architectures, not by the standard RGNN/TGNN message-passing class studied here. So the relevant comparison is not that $1$-RWL should match $2$-WL, but that it is the correct relational counterpart of the $1$-WL ceiling for our model class. We will add a brief discussion clarifying this point in the revision. Please also see our reply to reviewer 6rrp / Q2, where we address a closely related question.

---

> > ### Author Rebuttal · Reviewer_qnLq · 2026-04-04
> >
> > The authors fully resolved all my questions and concerns. I would like to raise my score to accept.

---

### Official Review · Reviewer_wg6V · 2026-03-12

**Soundness:** 3
**Presentation:** 2
**Significance:** 3
**Originality:** 3
**Overall Recommendation:** 5
**Confidence:** 4

**Summary:**

This paper extends the SELTH from standard GNNs (on static graphs) to relational and temporal GNNs. The main claim is that sufficiently wide sparse subnetworks can still preserve 1-RWL expressivity with positive probability under random pruning. The paper also argues that more expressive sparse subnetworks are easier to optimize.

**Compliance With Llm Reviewing Policy:**

Affirmed.

**Final Justification:**

The rebuttal addressed most of my comments and thus reinforced my first evaluation.

**Key Questions For Authors:**

Does the sparsity have any positive real-world impact, e.g., can we expect that this improves speed or memory of trained networks? Do we expect improved performance? In a fair setting, e.g., weight memory, should we rather train large networks pruned or smaller models unpruned to achieve lower loss? In particular, would this translate to real-world improvements, given that one usually requires structures sparsity.

Could you elaborate on this point please *"can increase the sparsity attainable at a given width for the same target RGNN. This also suggests a link to over-smoothing (Chuang et al.,
2025; Rusch et al., 2023), which can similarly reduce input separability."*? In particular, given that oversmoothing happens only in deeper layers, does it mean that deeper networks are more difficult to prune?

Also See above for more questios.

**Limitations:**

yes

**Strengths And Weaknesses:**

What I like:
- The topic is important: sparsity is useful only if pruning does not destroy the model distinguishing power. The main theoretical idea is interesting and ambitious: using a relational view to unify RGNNs, TGNNs, and XIMP/HIMP-style models.
- I liked the experiments because they are tied to the theory instead of being generic benchmark tables, e.g, Figure 1 is useful because it shows concretely how the bound depends on width, sparsity, branch count, and task complexity. Figure 2(a) is a good sanity check since it compares the theoretical probability bound with an empirical success probability on synthetic graphs.



Cons:
- The optimization claim is interesting, but Figure 2(b) does not fully disentangle expressivity from model size. Since subnetworks with lower sparsity naturally retain more parameters, it is unclear whether the observed improvement in gradient flow, loss, or accuracy comes from preserving task-relevant expressivity or simply from having larger subnetworks. A controlled comparison at fixed sparsity would make this claim much more convincing. E.g., using a fixed sparsity but different expressivity levels (through random sampling) and then comparing loss etc.
- The probability bound feels very wild / very conservative in practice. It depends on many quantities, and after worst-case aggregation and union bounds it becomes hard to know whether the derived probability is meaningful. I would really like to see one explicit numerical calculation in the main paper: take one actual experimental setup, plug in the numbers, and show what probability the theorem gives. I.e., the actual bound with all constant. Then, assuming the the bound is loose (which is ok), I would like to understand whether the bound at least gives the right ideas of how it correlates with the most important quantities, e.g., width or number of graphs.
- notation is dense and often overloaded. A concrete example: in Theorem 3.1, the paper first distinguishes branch and combine quantities, then later collapses them into a single worst-case quantity. This makes it hard to track what actually controls the bound.
Another concrete example: the TGNN to RGNN reduction around Lemma 3.3 is difficult to follow. It is not very clear, without rereading several times, where the temporal information is stored and how the relation-specific branches correspond to the original TGNN update. A tiny two-snapshot toy example would help a lot.
- The proof of Theorem 3.1 relies heavily on importing a lemma from prior work, but the transfer is not fully clear. I would suggest to make the proofs more self-contained to understand the step from static setting to the relational multi-branch setting.

---

> ### Author Rebuttal · Authors · 2026-03-30
>
> We thank the reviewer for the thorough and insightful review, for recognizing the importance of analyzing sparsity in this setting, and for noting that the main theoretical idea is interesting and ambitious. We appreciate the many constructive suggestions, which are very helpful for sharpening the presentation and further strengthening the paper. We address the individual points below.
>
> ## Comments
> **C1:** The optimization claim is interesting, but [...]
>
> **A:** This is a good suggestion. To follow up on it, we ran an additional *fixed-sparsity control experiment*: for four sparsity levels, we sampled 100 random masks with identical parameter counts and compared optimization behavior across different pre-training separability levels, using 30 training epochs. This directly isolates the effect of separability from mask size.
>
> The result strengthens our original conclusion: even at fixed sparsity, higher pre-training separability remains significantly associated with stronger gradient flow and lower final loss across all tested sparsity levels, and with higher accuracy at the higher sparsity levels. Thus, the optimization trend in Figure 2(b) is not merely a byproduct of larger subnetworks, but persists in a within-sparsity comparison as well.
>
> **C2:** The probability bound feels very wild / very conservative [...]
>
> **A:** The bound is designed as a conservative worst-case guarantee, but our current results already indicate that it captures the correct qualitative trends: Figure 1 shows the expected dependence on width, sparsity, $N_{\max}$, and $|\mathcal{B}|$, while Figure 2(a) compares the instantiated bound to the empirical mask success rate on a synthetic dataset.
>
> We plan to add one concrete experimental instantiation. Preliminary results from one XIMP configuration on a Polaris task and one local-TGNN configuration on tgbn-trade show that the bound is looser for XIMP than in the synthetic toy example of Figure 2a, but tighter for the local TGNN.
>
> We will also make clearer in the text that the main role of the bound is to capture the correct monotone dependence on the key quantities, rather than to serve as a tightly calibrated probability in every regime.
>
> **C3:** notation is dense and often overloaded [...]
>
> **A:** This is a helpful point. We agree that these parts can be presented more clearly, and we will revise the notation and explanation accordingly so that both the bound and the TGNN$\to$RGNN reduction are easier to follow.
>
> **C4:** The proof of Theorem 3.1 relies heavily on importing a lemma [...]
>
> **A:** The imported prior lemma is only one elementary ingredient, whereas the main contribution of Theorem 3.1 is its extension to the relational multi-branch RGNN setting. We will clarify this explicitly in the revision by summarizing the imported result at a high level in the main text and restating it more fully in the appendix, so that the inherited and novel parts are clearly separated.
>
> ## Questions
>
> **Q1:** Does the sparsity have any positive real-world impact, [...]
>
> **A:** Our main claim is not that unstructured sparsity automatically gives wall-clock speedups, but that sufficiently overparameterized RGNNs/TGNNs can contain sparse subnetworks that remain expressive and trainable.
>
> Sparsity can already reduce model size and weight storage, while runtime gains often depend on hardware/software support and may require structured or hardware-aware implementations. Our contribution is therefore a foundational expressivity/optimization result: substantial pruning need not destroy the distinctions required for learning, which supports sparsification in these richer architectural settings.
>
> Similarly, we do not claim that “large-then-prune” always outperforms a smaller dense model under every fair budget comparison; rather, our point is that pruning can still preserve task-relevant separability. We will clarify this practical scope in the revision.
>
> We also note that prior work has shown that unstructured sparsity can be leveraged in practice when supported by appropriate sparse kernels / hardware-aware implementations (e.g, Gale et al., Sparse GPU Kernels for Deep Learning, 2020).
>
> **Q2**: Could you elaborate on this point please [...]
>
> **A:** Yes, at fixed width the bound becomes less favorable as $L$ increases, since $\widetilde{\gamma}$ depends on $L$ through the exponent $L(M|\mathcal{B}|+1)$. Likewise, larger $\widetilde{s}_{\min}$ improves the bound, so if increased propagation depth is accompanied by reduced input separability, then both effects act in the same direction and make expressive pruning harder.
>
> At the same time, our theorem does not model over-smoothing directly; the connection is indirect, via its effect on separability. So the correct interpretation is not that deep networks are impossible to prune, but that, at fixed width, larger $L$ generally requires either stronger separation or more width to maintain the same target guarantee. We will clarify this point in the revision.

---

> > ### Author Rebuttal · Reviewer_wg6V · 2026-04-03
> >
> > Thanks for the detailed response. I increase my score.

---

### Decision · Program_Chairs · 2026-04-30

**Decision:**

Accept (spotlight)

**Comment:**

This work extends the Strong Expressive Lottery Ticket Hypothesis (SELTH), previously established for uni-relational GNNs, to the realm of multirelational and temporal GNNs, claiming that sufficiently wide sparse subnetworks of a GNN can preserve WL expressivity on static graphs. The authors prove theoretical guarantees supporting their claim and supplement them with experimental results.

Reviewers agree that this is a solid contribution. While the reviewers had some concerns around details such as real-world implications, the effect of L on the model, extensions to other activations, etc., many of these were adequately addressed during the rebuttal period.